# Vinculin recruitment to α-catenin halts the differentiation and maturation of enterocyte progenitors to maintain homeostasis of the *Drosophila* intestine

**Jerome Bohere, Buffy L Eldridge-Thomas, Golnar Kolahgar***

Department of Physiology, Development and Neuroscience, Downing St, University of Cambridge, Cambridge, United Kingdom

**Abstract** Mechanisms communicating changes in tissue stiffness and size are particularly relevant in the intestine because it is subject to constant mechanical stresses caused by peristalsis of its variable content. Using the *Drosophila* intestinal epithelium, we investigate the role of vinculin, one of the best characterised mechanoeffectors, which functions in both cadherin and integrin adhesion complexes. We discovered that vinculin regulates cell fate decisions, by preventing precocious activation and differentiation of intestinal progenitors into absorptive cells. It achieves this in concert with α-catenin at sites of cadherin adhesion, rather than as part of integrin function. Following asymmetric division of the stem cell into a stem cell and an enteroblast (EB), the two cells initially remain connected by adherens junctions, where vinculin is required, only on the EB side, to maintain the EB in a quiescent state and inhibit further divisions of the stem cell. By manipulating cell tension, we show that vinculin recruitment to adherens junction regulates EB activation and numbers. Consequently, removing vinculin results in an enlarged gut with improved resistance to starvation. Thus, mechanical regulation at the contact between stem cells and their progeny is used to control tissue cell number.

*For correspondence:
gk262@cam.ac.uk

**Competing interest:** The authors declare that no competing interests exist.

## Editor's evaluation

This study from Kolahgar and colleagues examines a potential mechanosensation mechanism in fly intestinal stem cells and their enteroblast progeny. The authors focus on vinculin, whose activity in other epithelial systems is regulated by mechanical tension. The manuscript's data clearly demonstrate a role for vinculin in suppressing the proliferation of midgut stem cells and the differentiation of their enteroblast progeny and suggest that this role is exerted specifically through enteroblast vinculin. The authors find that similar phenotypes are induced by genetic manipulations of vinculin, α-catenin, and myosin, and they argue that this similarity implies that vinculin activity in enteroblasts is mechanosensitive. These findings are potentially relevant to biologists interested in stem cells, tissue homeostasis, fate decisions, and mechanobiology.

## Introduction

Adult epithelial tissues are maintained through the timely production of correct specialised cells by resident stem cells, ensuring tissue function, and preventing disease. In the intestine for example, where the rate of tissue turnover is high, uncompensated cell loss can result in chronic inflammation, whereas over-proliferation and mis-differentiation can produce adenomas that constitute a sensitised background for tumour initiation (*Gehart and Clevers, 2019*). Hence, elucidating the

**eLife digest** Mechanical changes in the environment have recently emerged as important signals of cell division and production of specialised cell types. Exactly how these forces are sensed and contribute to this process in living tissues, however, remains unclear.

This question is particularly relevant in the lining of the gut. Endlessly exposed to intense mechanical stress and the passage of food, this tissue must constantly heal and renew itself. The intestinal cells that absorb nutrients from food, for example, are continually replaced as older cells are lost. This is made possible by immature 'progenitor' cells in the intestine dividing and maturing into various specialised cells – including fully functional absorptive cells – upon receiving the right mechanical and chemical signals. Errors in this carefully regulated process can result in too many or too few cells of the correct kind being produced, potentially leading to disease.

To explore how mechanical forces may help to control the renewal and maturation of new intestinal cells, Bohère et al. examined the role of vinculin in the guts of fruit flies (where cell fate decisions involve mechanisms largely similar to humans). Vinculin can regulate cell fate, sense mechanical forces, and interact with the complex structures that physically connect cells to each other.

Genetically altered flies that lacked vinculin had enlarged guts containing many more absorptive cells than those of normal flies, suggesting that the vinculin protein prevents over-production of these cells. Further experiments revealed that vinculin worked exclusively in the precursors of absorptive cells, keeping them in an immature state until new mature absorptive cells were required. This was achieved by vinculin acting upon – and potentially strengthening – the junctions connecting cells together. Finally, increasing the force within cells was shown to facilitate vinculin recruitment to these junctions.

This study clarifies the role that mechanical forces at the interface between cells play in controlling when and how intestinal progenitors mature in an organism. If these findings are confirmed in mammals, Bohère et al. hope that they could inform how tissues cope with the changing mechanical landscape associated with ageing and inflammation.

molecular mechanisms regulating cell renewal and fate acquisition will help further our understanding of fundamental rules governing tissue homeostasis and might open avenues for therapeutic intervention.

In the mammalian intestine, a combination of locally secreted and membrane-bound proteins regulates cell fate decisions across the crypt–villus axis through the activation of key signal transduction pathways, including Wnt and Notch (*Meran et al., 2017*; *Gehart and Clevers, 2019*). There is also an increased appreciation of a potential role of mechanical cues from surrounding cells – through cell–cell and cell–matrix adhesion protein complexes – in regulating cell fate determination, as observed in other systems (*Meran et al., 2017*; *Wickström and Niessen, 2018*). Although mechanosensing of the extracellular matrix is currently under investigation in intestinal organoid cultures (*Gjorevski et al., 2016*), the regulation of intestinal cell fate by adhesion complexes is in general poorly understood compared to canonical signalling pathways. At adherens junctions between neighbouring epithelial cells, transmembrane E-cadherin proteins engage in homophilic adhesion via their extracellular domain, and indirectly associate with the cytoskeleton by the recruitment of β-catenin via their intracellular domain, which can interact with various actin-binding proteins including α-catenin. Heterodimeric transmembrane proteins integrins and their associated proteins, including talin, contribute to focal adhesions at cell–matrix junctions, thus linking extracellular ligands to cytoskeletal proteins (*Hynes, 2002*; *Sun et al., 2016*).

In the intestine, as well as other mature renewing epithelia, an outstanding question is how mechanical forces are locally integrated and interpreted to fine-tune proliferation and differentiation. In *Drosophila* for example, genetic ablation of integrins in intestinal stem cells (ISCs) prevents overproliferation following over-activation of growth factor signalling pathways (*Lin et al., 2013*). However, why integrins are required for proliferation remains unsolved. Characterising *in vivo* the function of junction-associated intracellular proteins capable of responding to mechanical forces through change of conformation and activity might illuminate how mechanotransduction contributes to the regulation of tissue homeostasis.

One candidate of interest is the highly conserved actin-binding protein vinculin, as it is capable of stabilising both adherens junctions and focal adhesions under tension (*Bays and DeMali, 2017*) and of regulating cell fate decisions in cell culture models and *in vivo* in the mouse skin (*Holle et al., 2013*; *Kuroda et al., 2017*; *Wang et al., 2019*; *Biswas et al., 2021*). Vinculin protein consists of a head and a tail domain separated by a linker region. In cells, vinculin transits from a closed, auto-inhibited inactive conformation (whereby the head and tail domains interact with each other, preventing further protein–protein interactions), to an open, active conformation initiated by force-dependent protein unfolding (reviewed in *Carisey and Ballestrem, 2011*; *Bays and DeMali, 2017*), that enables the vinculin tail domain to bind F-actin. The head domain can bind v̲inculin-b̲inding s̲ites (VBS) present in other proteins (including α-catenin and talin). Notably, intracellular tension induces conformational changes in α-catenin and talin, exposing their cryptic VBS and initiating vinculin binding (*del Rio et al., 2009*; *Yonemura et al., 2010*; *Yao et al., 2014*). This allows a transition of vinculin into an open conformation able to bind to the actin cytoskeleton, in turn enabling adhesion complexes to withstand higher forces in cells under tension (*le Duc et al., 2010*; *Dumbauld et al., 2013*; *Thomas et al., 2013*). Thus, vinculin is one of the best characterised mechanoeffectors, as mechanical force on both talin and α-catenin is converted into differential vinculin recruitment.

*In vivo*, loss of *vinculin* is embryonic lethal in mice due to defects in neural tube closure and heart development (*Xu et al., 1998*). In *Drosophila*, loss of *vinculin* is homozygous viable and fertile (*Alatortsev et al., 1997*; *Klapholz et al., 2015*), precluding an essential role during embryonic development. In post-embryonic muscles, *Drosophila* vinculin regulates actin organisation at integrin junctions (*Bharadwaj et al., 2013*; *Green et al., 2018*), and is lethal if expressed in a constitutively active form (*Maartens et al., 2016*).

In epithelia, *Drosophila* vinculin contributes to myosin II mediated tension sensing at cell–cell junctions through its interaction with α-catenin (*Case et al., 2015*; *Jurado et al., 2016*; *Kale et al., 2018*; *Alégot et al., 2019*), with functions at integrin adhesion sites in epithelia remaining to be identified. Conditional *vinculin* knock-out in the mouse skin epidermis revealed a dispensable role in reinforcing newly formed adherens junctions in the stratified epidermis (*Rübsam et al., 2017*), and a more prominent role in maintaining stem cells in a quiescent state via stabilisation of adherens junctions and contact inhibition in the hair follicle bulge (*Biswas et al., 2021*). The variety of phenotypes observed so far among different tissues and organisms suggest that vinculin functions in a highly context-specific manner, and thus, characterising its contribution to tissue homeostasis in multiple systems is required to improve our understanding of mechanotransduction.

The role of vinculin in epithelial maintenance and stem cell lineage decisions is unknown in the intestine. Here, we investigate *in vivo* the role of vinculin in the maintenance of the *Drosophila* intestine, a relatively simple model of homeostatic adult epithelium used for its genetic tractability (reviewed in *Miguel-Aliaga et al., 2018*). In this system, resident stem cells are scattered among differentiated absorptive and secretory cells and respond to conserved signalling transduction pathways to, respectively, divide and differentiate. Upon cell division, ISCs self-renew and produce precursor cells able to differentiate into either secretory enteroendocrine cells (EEs) or large absorptive enterocytes (ECs) (*Micchelli and Perrimon, 2006*; *Ohlstein and Spradling, 2006*). *Drosophila* ISCs and their undifferentiated daughter cells, enteroblasts (EBs), are collectively called progenitor cells and express the transcription factor *escargot* (*esg*) (*Figure 1A*). As EBs differentiate into ECs, through changes in gene expression, cells develop membrane protrusions and adopt a migratory phenotype, allowing them to move between cells; EBs also progressively increase in cell volume, resulting in an extended contact area on the basal side (*Antonello et al., 2015*; *Rojas Villa et al., 2019*). In the *Drosophila* intestine, loss of integrin-mediated adhesion has deleterious effects on stem cell maintenance and proliferation (*Goulas et al., 2012*; *Lin et al., 2013*; *Okumura et al., 2014*; *You et al., 2014*) and E-cadherin-mediated adhesion regulates stem cell maintenance and differentiation (*Choi et al., 2011*; *Maeda et al., 2008*; *Zhai et al., 2017*).

We show that vinculin regulates differentiation into the absorptive cell lineage, independently of integrins at cell–matrix sites and through its association with α-catenin at cell–cell junctions.

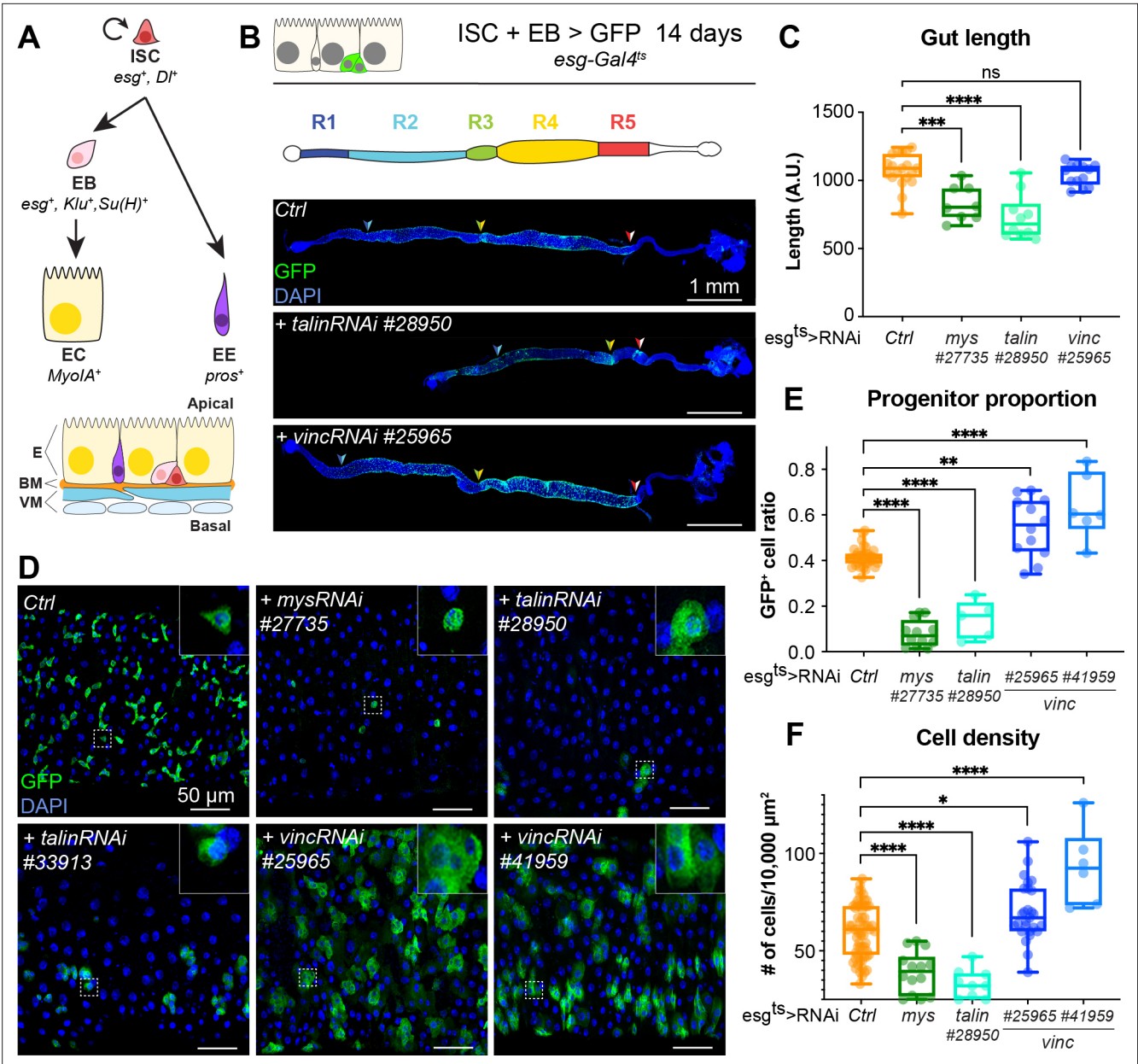

**Figure 1.** *talin* and *vinculin* knockdowns produce opposite phenotypes in the gut. (**A**) Lineage of the adult midgut: intestinal stem cells (ISCs) self-renew and give rise to post-mitotic enteroblasts (EBs) which terminally differentiate into enterocytes (ECs), and enteroendocrine cells (EEs). Cell type-specific genes are shown in italics. Schematic at bottom shows overall tissue organisation. E: epithelium, BM: basement membrane, VM: visceral muscles. (**B, C**) Gut regions R4 and R5 are reduced by RNAi knockdown of *talin* but not *vinculin*, green fluorescent protein (GFP; green) marks cells expressing the RNAi, and nuclei are blue (DAPI, 4',6-diamidino-2-phenylindole). Anterior is to the left in this and all subsequent figures. (**D, E**) Surface view of region R4/5. RNAi knockdown of *integrin* and *talin* shows reduced number of RNAi-expressing ISC/EB cells (GFP+) and their rounded morphology (insets). In contrast there are more ISC/EB cells in the absence of vinculin and they are enlarged. These changes result in an overall change in total cell density (**F**). Two-tailed Mann–Whitney tests were used: ns: not significant; ****p < 0.0001, ***p < 0.001, **p < 0.01, *p < 0.05.

The online version of this article includes the following source data and figure supplement(s) for figure 1:

**Source data 1.** Raw data for *Figure 1*, *Figure 1—figure supplement 1*.

**Figure supplement 1.** Vinculin localises to cell–cell and cell–extracellular matrix junctions and affects intestinal cell divisions.

## Results

### Vinculin has opposite effects on progenitor numbers and cell density compared to integrin and talin

In the *Drosophila* gut, complete loss of function of the ubiquitous β-PS integrin subunit (*myospheroid*, *mys*) progressively induces ISC loss (*Lin et al., 2013*). With the goal of uncovering intracellular factors mediating integrin function in stem cells, we investigated the role of two integrin-associated mechanosensing proteins, talin and vinculin (*Atherton et al., 2016*; *Klapholz and Brown, 2017*). Talin is important for the establishment of EC polarity (*Chen et al., 2018*) and for stem cell maintenance (*Lin et al., 2013*). The role of vinculin, which is expressed ubiquitously in the gut and localises to cell–matrix and cell–cell junctions (*Figure 1—figure supplement 1A–F*), has not been characterised in the adult intestine. We used RNA interference to downregulate β-PS integrin, talin and vinculin in *esg*-expressing progenitor cells visualised with green fluorescent protein (GFP). Downregulation of β-PS integrin and talin led to gut shortening in the posterior region (*Figure 1B, C*) associated with loss of progenitors (see reduction in GFP$^+$ cells in *Figure 1D, E*) adopting a circular shape in contrast with control cells that displayed a triangular shape (insets, *Figure 1D*). Expression of two different validated RNAi lines against *vinculin* (*vinc*) did not impact gut length and instead led to an increase in GFP$^+$ cells number and total cell density (*Figure 1B–F*), associated with increased proliferation (*Figure 1—figure supplement 1G*). No rounding of the progenitor cells was observed, instead cells appeared larger suggestive of accelerated differentiation (insets *Figure 1D*). These opposite phenotypes suggested that vinculin regulates progenitor production and tissue size, possibly independently of integrin/talin adhesion and prompted us to clarify the role of vinculin in intestinal tissue maintenance.

### Intestinal cell production is increased upon global loss of *vinculin* function

To study the impact of *vinc* complete loss of function on guts, we used the null allele *vinc$^{102.1}$*, a deletion removing approximately 30 kb of genomic DNA including the entire *vinc* coding sequence (*Klapholz et al., 2015*). After dissection, we noted that the intestines of *vinc$^{102.1}$* mutants were wider compared to control *yw* flies, despite being of comparable lengths. We tested if the difference in gut width reflected a gross difference in body size by measuring the adult wing surface areas of *yw* and *vinc$^{102.1}$* flies. As we found them to be comparable (*Figure 2—figure supplement 1*) and in accordance with (*Sarpal et al., 2019*), we next set out to establish if the increased gut width resulted from an increase in cell numbers. Guts were processed for electron microscopy (*Figure 2A*) or immunofluorescence (*Figure 2B*) and cells were counted in cross-sections. Both methods showed increased cell numbers in *vinc$^{102.1}$* posterior midguts (see quantification of DAPI$^+$ nuclei from cross-sections of fluorescently labelled guts in *Figure 2B*). Importantly, even when we compared guts of similar width (between 190 and 220 µm in *Figure 2B*), we found that *vinc$^{102.1}$* gut cross-sections had approximately twice as many cells as *yw* guts.

To characterise how vinculin regulates tissue homeostasis, we genetically labelled all intestinal progenitor cells (*esg$^+$*) and their progeny with GFP (*Figure 2C–E*), combining the labelling cassette *esg$^{ts}$F/O* (or 'escargot-Flip-Out', *Jiang et al., 2009*) in *yw* or *vinc$^{102.1}$* male flies (see *Figure 2—figure supplement 2* and Methods for experimental details). We compared the extent of GFP labelling as a proxy for cell production over a period of 14 days post induction. During this period, moderate tissue turnover occurs in control *yw* guts (*Jiang et al., 2009*), as evidenced by groups of more than two cells that included some ECs, recognised by their large polyploid nuclei (*Figure 2D, E*, *yw* guts). Whereas the distribution of newly produced cells was largely uniform and low across the posterior midguts of *yw* flies (*Figure 2D*, top panel), guts of *vinc$^{102.1}$* flies presented with a strong labelling at the interface between R4/5 regions of the posterior midgut (regions defined in *Buchon et al., 2013*; *Figure 2D* middle panel, compare to cartoon *Figure 1B*). Quantification of GFP signal intensity (as described in *Figure 2—figure supplement 2B*) across 27 guts showed a penetrant phenotype (see heatmaps *Figure 2D*). The phenotype was fully rescued in flies carrying a genomic *vinc* rescue construct (Vinc-RFP, *Klapholz et al., 2015*) (*n* = 34, *Figure 2D*, bottom panel), confirming that it is *vinc* loss that accelerates tissue turnover. This was confirmed by measuring gut width (*Figure 2F*) and cell density (*Figure 2G*), which both reflected increased cell production. Altogether, the increase in cells

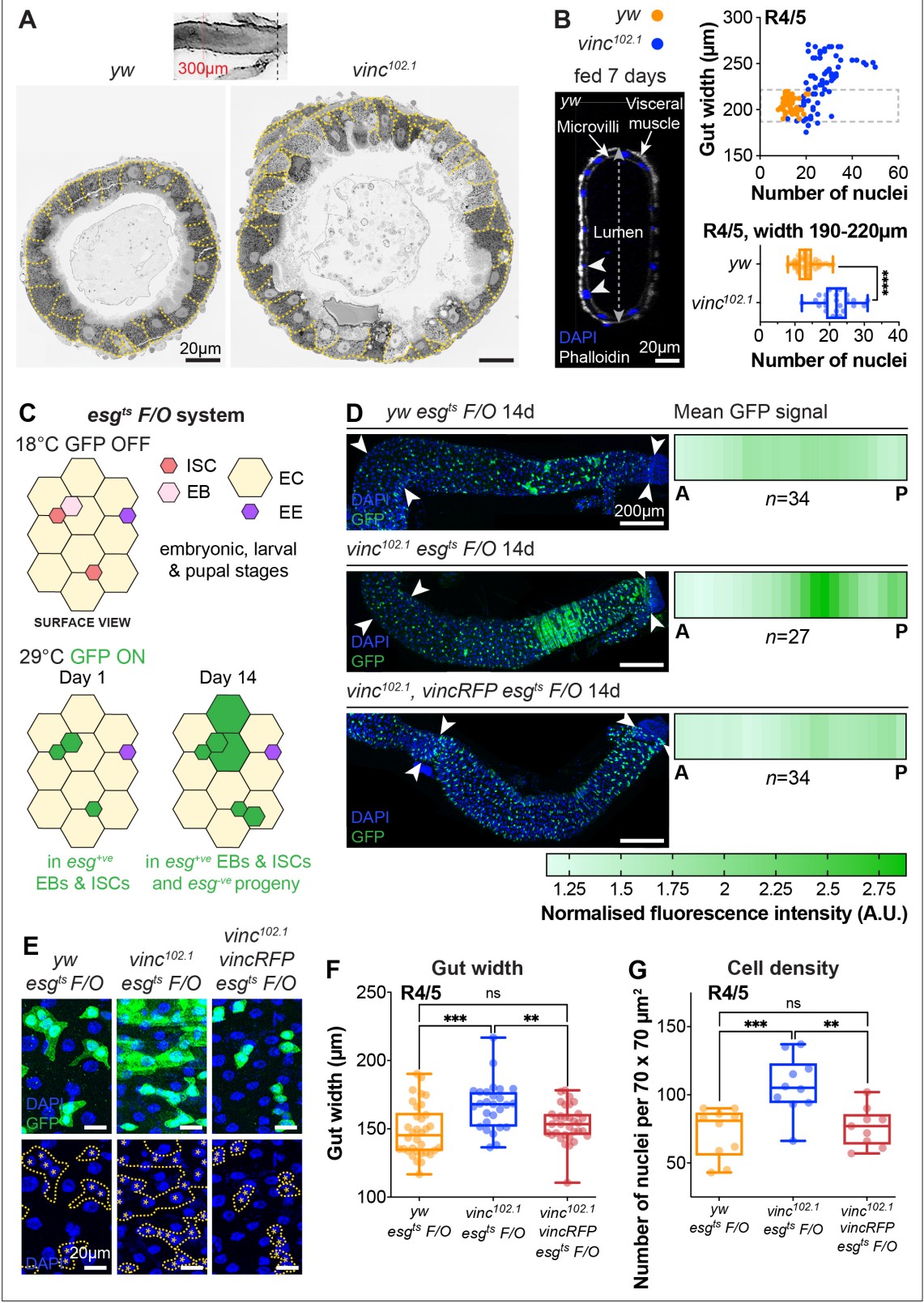

**Figure 2.** Global loss of vinculin accelerates intestinal cell production. (**A**) SEM cross-sections of *yw* and *vinc*[102.1] midguts. Dashed lines indicate cell boundaries. *vinc*[102.1] guts are wider than *yw* guts. Inset indicates that cross-sections were taken 300 μm anterior to the Malpighian tubule attachment sites. (**B**) Confocal cross-sections of midguts stained with Phalloidin (white) and DAPI (blue) were analysed to extract gut width (arrow in lumen) and epithelial cell numbers (arrowheads indicate nuclei of epithelial cells). Nuclei number per gut cross-section were plotted against R4/5 midgut width. *yw:*

*Figure 2 continued on next page*

*Figure 2 continued*

n = 55 datapoints from five guts, *vinc*[102.1]: n = 77 datapoints from seven guts. *vinc*[102.1] midguts contain more cells than *yw*, even when comparing guts of the same width. (subset of guts with comparable width [dashed box] replotted in bottom graph *yw*: n = 55 datapoints from five guts, *vinc*[102.1]: n = 34 datapoints from four guts). (**C**) Top view schematic of the *esg*[ts] F/O system. At the permissive temperature for Gal4 expression (29°C), the intestinal stem cell (ISC)- and enteroblast (EB)-specific *esg-Gal4* drives expression of both *UAS-GFP* and *UAS-flp*, which mediates permanent, heritable expression of green fluorescent protein (GFP). Thus, as ISCs divide and EBs differentiate during adulthood, all cells which arise from progenitors will express GFP. (**D**) Left panels: progenitors and newly produced cells (GFP+, green) 14 days after induction of Gal4 expression. A large GFP+ region is present in the R4/5 region of *vinc*[102.1] *esg*[ts] F/O guts. Arrowheads indicate the region along which heatmaps were generated. Images are *z*-projections through half the gut depth. Right panels: heatmaps of GFP-fluorescence intensity. Dark green corresponds to high levels of GFP fluorescence, indicating elevated tissue turnover. *yw esg*[ts] F/O n = 34 guts, *vinc*[102.1] *esg*[ts] F/O n = 27 guts, *vinc*[102.1] *vincRFP esg*[ts] F/O n = 34 guts, from three replicates. (**E**) *esg*[ts] F/O patches with GFP-DAPI overlay and DAPI only. Dashed lines indicate patch boundaries. Asterisks indicate GFP+ nuclei. Images are *z*-projections. (**F**) Quantification of gut width. *yw esg*[ts] F/O n = 34 guts, *vinc*[102.1] *esg*[ts] F/O n = 27 guts, *vinc*[102.1] *vincRFP esg*[ts] F/O n = 34 guts. (**G**) Quantification of cell density. *yw esg*[ts] F/O n = 10 guts, *vinc*[102.1] *esg*[ts] F/O n = 10 guts, *vinc*[102.1] *vincRFP esg*[ts] F/O n = 9 guts. Two-tailed Mann-Whitney tests were used: ns: not significant, **** $p < 0.0001$, *** $p < 0.001$, ** $p < 0.01$.

The online version of this article includes the following source data and figure supplement(s) for figure 2:

**Source data 1.** Raw data for *Figure 2*, *Figure 2—figure supplement 1*.

**Figure supplement 1.** Adult wing size is not affected by loss of vinculin.

**Figure supplement 2.** Quantification of tissue turnover.

observed when vinculin is removed indicates that vinculin inhibits cell renewal under basal conditions (locally and possibly through tissue interactions).

## Vinculin is not required in ISCs to regulate proliferation

As *Drosophila* intestinal cell production relies solely on ISCs (*Micchelli and Perrimon, 2006*; *Ohlstein and Spradling, 2006*), we speculated that vinculin might negatively regulate stem cell proliferation, in a similar fashion as the recently observed control of bulge stem cell proliferation in mice hair follicle (*Biswas et al., 2021*). We expressed *vincRNAi* and YFP in all ISCs and monitored the stem cell pool and tissue size after 14 and 28 days of expression at permissive temperature (*Figure 3*). Unlike *vinc* knockdown in both ISCs and EBs, we did not detect any change in stem cell proportion or tissue density (*Figure 3B, C*). This was again in contrast to *mys* or *talin* knockdown that led to a clear rounding and detachment of ISCs from the basement membrane as seen by the delocalisation of β-catenin to the basal side of the cell (*Figure 3A*, orthogonal views), possibly explaining the significant reduction of the stem cell population (*Figure 3B*) and cell density (*Figure 3C*) not observed with ISC-specific *vinc* knockdown. This suggested to us that vinculin is required in EBs to maintain tissue homeostasis.

## The pool of EC progenitors expands upon vinculin depletion

To explore the role of vinculin in EBs, we examined *vinc* mutant cells. In the *Drosophila* intestine, EBs are predominantly produced after asymmetric stem cell divisions, but can in fewer instances result from symmetric divisions failing to retain the ISC, and instead producing two EBs (*O'Brien et al., 2011*; *de Navascués et al., 2012*). As Notch is activated in EBs (*Figure 1A*; *Micchelli and Perrimon, 2006*; *Ohlstein and Spradling, 2006*; *Ohlstein and Spradling, 2007*), we combined *yw* or *vinc*[102.1] males to a Notch Responsive Element fused to lacZ (*Furriols and Bray, 2001*) and immunostained dissected guts for β-galactosidase and Dl to quantify the relative proportions of EBs and ISCs, respectively (*Figure 4A–C*). *vinc*[102.1] guts had a clear accumulation of NRE-lacZ+ cells (especially in the region of high turnover R4/5 marked with longitudinal arrows in the *vinc*[102.1] gut in *Figure 4A*), and the overall proportion of stem cells was reduced (*Figure 4C*). We had a similar observation using a transcriptional reporter of the JAK–STAT signalling pathway, which is also required during EC differentiation (*Beebe et al., 2010*; *Figure 4—figure supplement 1*).

It was somewhat surprising to observe cells with active Notch signalling accumulating in a context where the ligand-presenting cells (Dl+ ISCs) were sparse. One possible explanation would be that instead of dividing predominantly asymmetrically to maintain one stem cell and produce one daughter EB cell with Notch signalling active, as described in homeostasis (*O'Brien et al., 2011*; *de Navascués et al., 2012*; *Guisoni et al., 2017*; *Hu and Jasper, 2019*), ISCs may more frequently divide symmetrically in *vinc*[102.1] tissues, producing two EBs with Notch signalling active. Another possibility would be that EBs continuously signal to ISCs to divide more: this way, even a relatively smaller number of

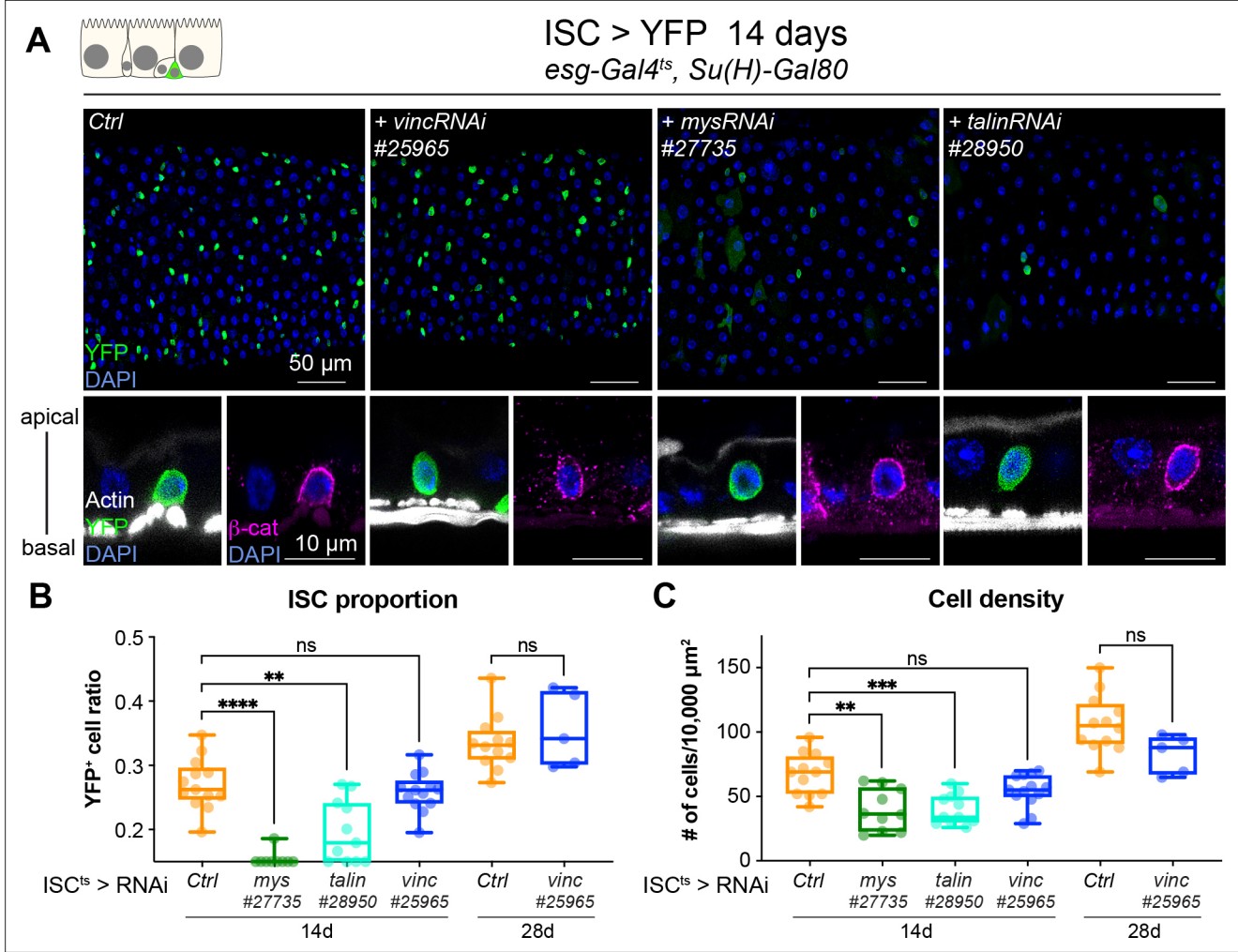

**Figure 3.** Vinculin is not required in stem cells to regulate cell proliferation. (**A**) Fourteen-day RNAi-mediated knockdown of *mys*, *talin*, and *vinculin* in intestinal stem cells (ISCs) (YFP, green). Nuclei are blue throughout the figure (DAPI). *mys* and *talin*, but not *vinc*, knockdown induce stem cell rounding and detachment from the basal side orthogonal views from guts stained with Phalloidin (white) and β-catenin (β-cat, magenta). Quantification of the ratio of YFP+ ISCs to the total number of cells (**B**) or cell density (**C**), after 14 or 28 days of RNAi expression. Two-tailed Mann-Whitney tests were used: ns: not significant, **** p<0.0001, *** p<0.001, ** p<0.01.

The online version of this article includes the following source data for figure 3:

**Source data 1.** Raw data for *Figure 3*.

ISCs in the tissue would produce more differentiating cells. This type of feedback has been described for EBs undergoing maturation, thus we sought to identify maturing EBs. Under normal conditions, diploid EBs remain dormant after production, until in response to tissue demand, they activate signalling programmes leading to their maturation, migration, endoreplication, and differentiation into ECs (*Antonello et al., 2015*; *Chen et al., 2016*; *Choi et al., 2011*; *Rojas Villa et al., 2019*; *Xiang et al., 2017*; *Zhai et al., 2017*). To distinguish between dormant and activated EBs, we measured the nuclear size of ISCs (Dl+), EBs (*NRE-lacZ*+), and ECs (Dl− *NRE-lacZ*− and polyploid), as a proxy of the degree of differentiation, which involves cells becoming increasingly polyploid (*Figure 4D*). In control tissues, EBs were distributed in two groups (see *yw* violin plots, *Figure 4D*), with dormant and activated cells corresponding to smaller and bigger nuclei/cells, respectively (*Antonello et al., 2015*; *Rojas Villa et al., 2019*). In contrast, the large majority of *vinc102.1* EBs had bigger nuclei, suggesting reduced numbers of cells were in the dormant stage in the absence of *vinc*. We also noted that most ECs in *vinc102.1* guts had smaller nuclei compared to *yw* guts, suggesting incomplete maturation of ECs (*Figure 4D*).

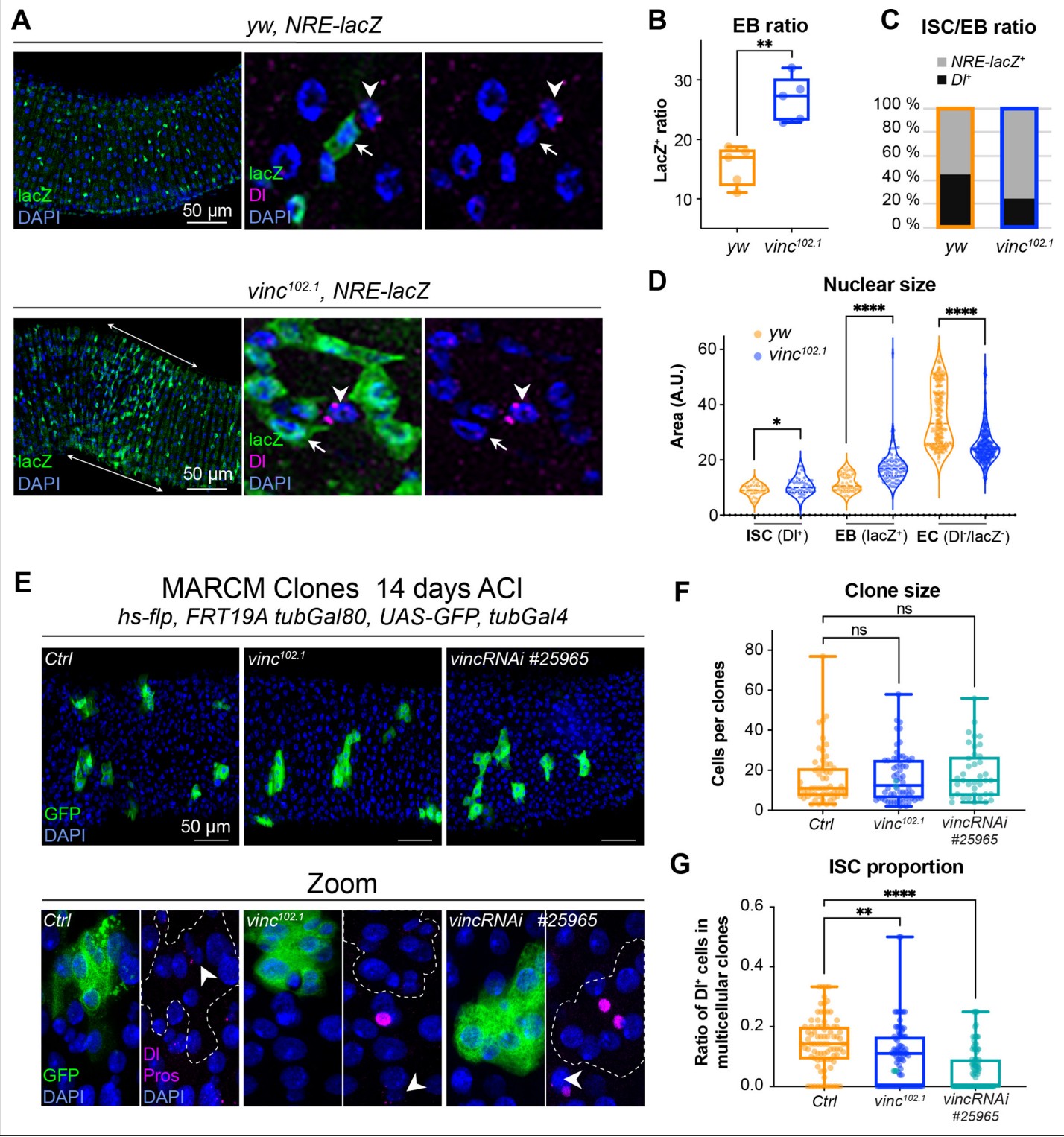

**Figure 4.** The enteroblast (EB) pool expands upon vinculin depletion. (**A**) EBs in 14-day-old *yw* or *vinc^102.1* guts of male flies, marked by expression of Notch reporter NRE-lacZ (β-galactosidase antibody staining green, arrows in insets). Dl marks intestinal stem cells (ISCs) (magenta, arrowheads). Nuclei are blue throughout the figure (DAPI). Arrows highlight the region R4/5 where NRE-lacZ+ cells accumulate in *vinc^102.1* guts. (**B**) Quantification of EB ratio (number of NRE-lacZ+ cells/total number of cells) showing the excess of EBs in *vinc^102.1* guts. (**C**) Quantification of ISC/EB ratio. (**D**) Quantification of nuclear size of the different cell populations. Note the distribution of EBs in two groups in *yw* guts, representing dormant and activated EBs. *vinc^102.1* EBs have overall larger nuclear size. (**E**) Green fluorescent protein (GFP)-labelled mitotic clones representing wild-type (left), *vinc^102.1* homozygous mutant (middle) or *vincRNAi*-expressing cells (right). In all conditions, unlabelled cells are considered wild-type. Zoom panels show clones (GFP, green) outlined

*Figure 4 continued on next page*

*Figure 4 continued*

with dotted lines, ISCs (anti-Dl, arrowheads, magenta), and enteroendocrine cells (EEs) (anti-Pros, magenta nuclear signal). Examples of multicellular clones devoid of Dl staining are shown for *vinc*[102.1] and *vincRNAi*. (**F**) Quantification of cell numbers in clones containing Dl[+] cells. Ctrl: *n* = 56 clones across 26 guts; *vinc*[102.1]: *n* = 58 clones across 34 guts; *VincRNAi*: *n* = 40 clones across 25 guts. (**G**) Quantification of stem cell proportion in multicellular clones (number of Dl[+] cells/number of cells per clone). Ctrl: *n* = 85 clones across 26 guts; *vinc*[102.1]: *n* = 86 clones across 34 guts; *VincRNAi*: *n* = 92 clones across 25 guts. Two-tailed Mann-Whitney tests were used: ns: not significant, **** p<0.0001, ** p<0.01.

The online version of this article includes the following source data and figure supplement(s) for figure 4:

**Source data 1.** Raw data for *Figure 4* and *Figure 4—figure supplement 1*.

**Figure supplement 1.** The number of cells with active JAK–STAT signalling increases in *vinc*[102.1] guts.

**Figure supplement 2.** Clones lacking vinculin accumulate enteroblasts (EBs) but not enteroendocrine cells (EEs).

These cellular behaviours could be due to global changes for example resulting from tissue interactions in homozygous mutant flies, therefore, we generated GFP-labelled control, homozygous *vinc*[102.1] mutant or *vinc*-RNAi-expressing mitotic clones derived from a single stem cell and compared the size and composition of multicellular clones. We did not detect differences in overall clone size, indicating that similar numbers of cells were produced from a single stem cell (*Figure 4F*). We noted however that the proportion of Dl[+] ISCs was reduced in *vinc*[102.1] mutant or *vinc*-RNAi-expressing clones (*Figure 4E, F*), with some clones devoid of Dl staining (see Zooms in *Figure 4E*). Thus, despite having fewer stem cells in the absence of vinculin, clones can grow to comparable sizes to control clones, indicating stem cells must have divided more frequently. The presence of clones devoid of Dl[+] stem cells supports the possibility that mitotic events producing two differentiated daughter cells may occur more frequently in the absence of vinculin, as suggested from data in *vinc* homozygous tissues. As the clones contained comparable proportions of EEs (*Figure 4—figure supplement 2A*), but higher ratios of cells with bigger nuclei (indicative of EB maturation, *Figure 4—figure supplement 2B*), it suggested that vinculin might control EB numbers, at the expense of the ISC population.

Together with our previous results showing that stem cell divisions increase upon *vinc* knockdown in ISCs+EBs (*Figure 1—figure supplement 1G*), but that *vinc* knockdown in ISCs had no effect (*Figure 3*), we hypothesised that vinculin in EB cell autonomously controls differentiation, and non-cell autonomously regulate ISC proliferation.

## Vinculin maintains EBs in a quiescent state

To evaluate the role of vinculin in EBs, we compared loss of vinculin in both ISCs and EBs (*esg*[+] > *VincRNAi*) versus EBs only (*klu*[+] > *VincRNAi*, **Korzelius et al., 2019**; **Reiff et al., 2019**). We monitored EB differentiation by expressing under the control of the same promoter two fluorophores, GFP and RFP, with differing protein half-lives (**Antonello et al., 2015**), such that newly differentiated EBs can be recognised by the absence of GFP and presence of nuclear RFP (see methods and cartoons in *Figure 5A, D*). *vincRNAi* expression in *esg*[+] cells led to faster progenitor differentiation cells (*Figure 5A–C*; note the accumulation of GFP[−]RFP[+] cells) and an increased differentiation index calculated as the ratio of GFP[−]RFP[+] cells/total cells (*Figure 5C*). Expression of *vincRNAi* in *klu*[+] cells confirmed a role of vinculin in EBs to control their production (*Figure 5—figure supplement 1D*) and to indirectly regulate proliferation (*Figure 5—figure supplement 1A–C*) and cell density (*Figure 5—figure supplement 1E*). This is reminiscent of the accumulation of activated EBs and newborn ECs observed in *vinc*[102.1] guts. It should be noted however that although GFP[−]RFP[+] cells with bigger nuclei were frequently observed following *vincRNAi* expression (see zooms, right panels in *Figure 5E*), the differentiation index was not significantly different to control guts (*Figure 5F*), possibly because of differences in Gal4 expression, or because terminal EC differentiation was affected.

Thus, loss of vinculin in EBs feeds back on ISCs, stimulating them to divide. Occasionally, we observed instances of mitotic EBs (PH3[+] Klu[+] cells) following *vincRNAi* expression (*Figure 5—figure supplement 1B*). This abnormal occurrence of mitotic marker in EB cells, which are normally post-mitotic, suggests these newly generated EBs have kept some mitotic potential. Indeed, these cells are reminiscent of the division capable EB-like cells previously described (**Kohlmaier et al., 2015**). In contrast to vinculin knockdown in EBs (*Figure 5—figure supplement 1E*), knockdown in ECs did not affect cell density (*Figure 5—figure supplement 2*), suggesting that vinculin acts specifically in EBs to maintain them in a dormant state to tune differentiation rates to tissue needs.

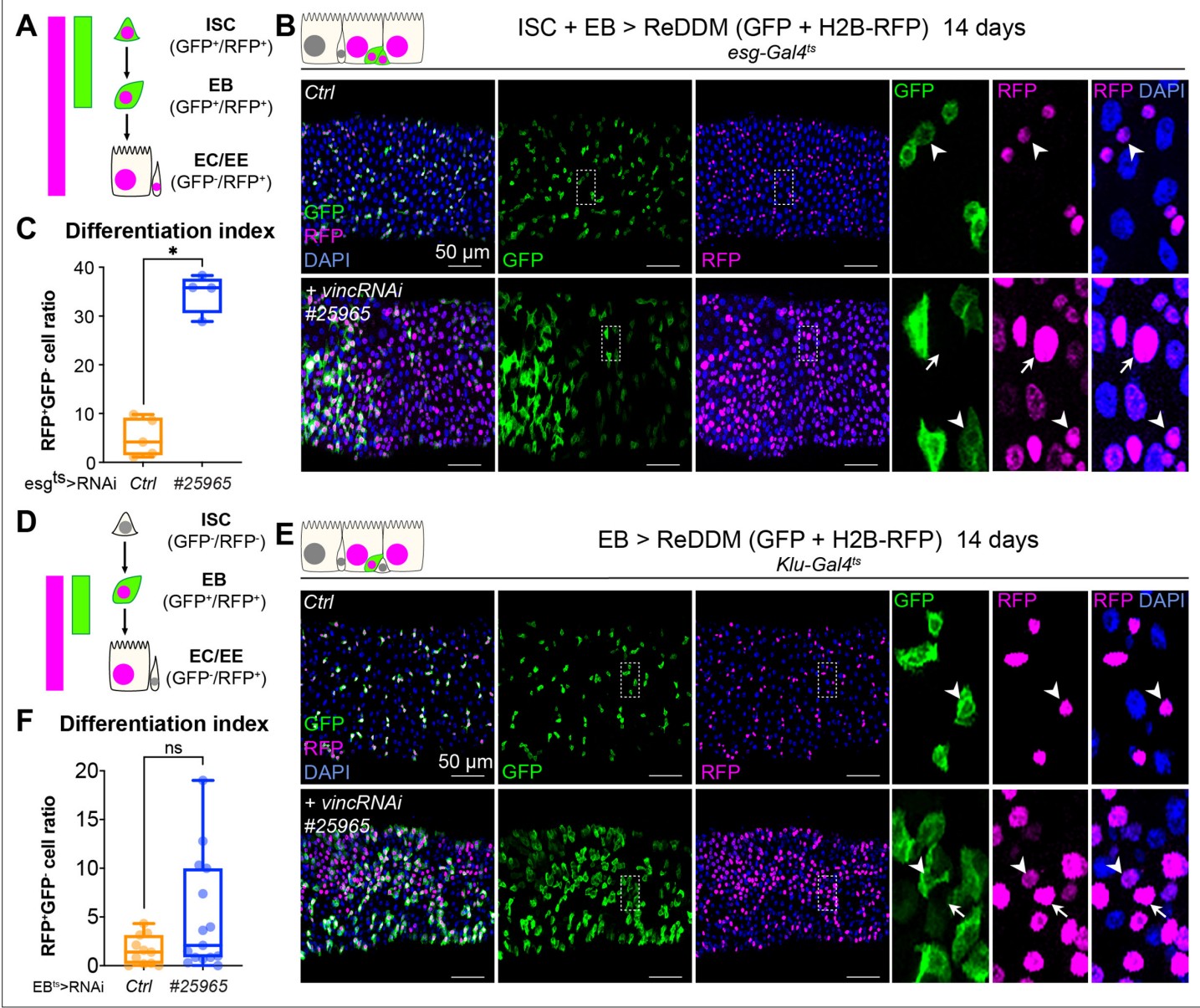

**Figure 5.** Vinculin slows enterocyte precursor differentiation. (**A**) Schematic of the ReDDM system expressed in progenitor cells. (**B**) Expression of ReDDM in progenitors with *vincRNAi#25965* or without (Ctrl) for 14 days. Close-ups (dashed line rectangles) show GFP−RFP+ differentiated cells (arrows) upon *vinc* knockdown. Arrowheads indicate GFP+RFP+ progenitors. (**C**) Quantification of the differentiation index (proportion of GFP−RFP+ cells) Expression of *vincRNAi#25965* in progenitors accelerates differentiation. (**D**) Schematic of the ReDDM system expressed in enteroblasts (EBs). (**E**) *vincRNAi#25965* expression in EBs leads to the accumulation of GFP+ cells. Close-ups show GFP+RFP+ (arrowheads) and GFP−RFP+ (arrows) cells seen more in *vincRNAi#25965*-expressing guts. (**F**) Quantification of the differentiation index when RedDM is expressed with *Klu-Gal4*. No statistical difference was noted when *vincRNAi#25965* was expressed, suggesting Klu+ cells accumulate and stall in this condition. Two-tailed Mann-Whitney tests were used: ns: not significant, * p<0.05.

The online version of this article includes the following source data and figure supplement(s) for figure 5:

**Source data 1.** Raw data for *Figure 5* and *Figure 5—figure supplements 1 and 2*.

**Figure supplement 1.** Vinculin knockdown in enteroblasts (EBs) results in increased proliferation.

**Figure supplement 2.** Vinculin knockdown in enterocytes (ECs) does not affect cell density.

## Vinculin intestinal function is mediated by its association with α-catenin

Adherens junctions between dormant EBs and ISCs regulate ISC proliferation (*Choi et al., 2011*; *Zhai et al., 2017*). Therefore, a function of vinculin that would explain its role in maintaining EB quiescence and reduced ISC proliferation might be via regulation of these junctions. We found that vinculin was present in adherens junctions, co-localising with β-catenin (*Figure 1—figure supplement 1F*).

If the effects of loss of vinculin in EBs are via its binding to α-catenin, then we may expect similar defects when α-catenin is removed. We therefore examined the phenotype of *α-catenin (α-cat)* down-regulation in *esg⁺* cells (*Figure 6A–C*). Nuclear size quantification of GFP⁺ progenitor cells showed that as in *vincRNAi*-expressing guts, *α-cat* RNAi expression resulted in larger *esg⁺* cells, equivalent in size to differentiating EBs and ECs (*Figure 6B*) and an overall accumulation of progenitors (*Figure 6C*). Consistent with being downstream of α-catenin, knockdown of *vinc* did not alter α-catenin membrane localisation (*Figure 6—figure supplement 1A*), but knockdown of α-catenin reduced vinculin recruitment to adherens junctions (*Figure 6—figure supplement 1B, C*). Thus, α-catenin is required upstream of vinculin in progenitor cells to regulate tissue turnover.

To test whether vinculin function was mediated by its ability to bind to α-catenin via the VBS domain, we downregulated *α-cat* as above and concomitantly expressed RNAi-resistant α-catenin constructs containing or lacking domains impacting vinculin binding (*Alégot et al., 2019*; *Figure 6D*). As before we used the ReDDM lineage timer to measure the degree of differentiation (*Figure 6E*). Guts rescued by co-expression of full-length α-catenin (α-Cat FL) were indistinguishable from wild-type controls (expression of ReDDM only in *esg⁺*), with similar rates of differentiation (*Figure 6E–G*). In contrast, co-expression of α-catenin constructs with the VBS deleted (α-Cat ΔM1a and ΔM1b), failed to rescue EBs accumulation (although intriguingly, we noted a milder phenotype with α-Cat ΔM1b). Deleting the adjacent M2 domain did not impair the ability of α-catenin to rescue this phenotype (*Figure 6E–G*). We next tested the role of the vinculin/α-catenin interaction in ISCs or EBs by co-expressing *α-Cat RNAi* and α-Cat ΔM1a and observed that the absence of VBS in α-catenin has no effect in ISCs but promotes EB differentiation and increase in numbers (*Figure 6—figure supplement 2*). We conclude that vinculin represses activation of a pool of EC precursors via its interaction with α-catenin. This suggests a model where vinculin, in its open, active form, contributes to keeping cellular junctions of EBs under tension, and this keeps the cells in a dormant state. In the absence of vinculin (or as the protein switches back to a closed, inactive conformation), EB junctions may destabilise faster, facilitating the transition from a dormant to an activated, migratory EB state and further differentiation into ECs. In support of this model, EB-specific knockdown of E-cadherin (*Drosophila shotgun*, *shg*) and α-catenin triggered faster differentiation (shown after 3 days of RNAi expression in *Figure 6—figure supplement 3*, as longer expression with Klu-Gal4 compromised fly survival). Since vinculin could in principle have a role at both cell–cell and cell–matrix adhesion sites (*Figure 1—figure supplement 1E, F*), we also tested the role of integrin-mediated adhesion in EBs and found that integrin and talin also contribute to EB differentiation (*Figure 6—figure supplement 4*). Thus, both adhesion machineries are required in the EB–EC differentiation step, with a clear role for vinculin/α-catenin in EBs only, and additional roles for integrin/talin in ISC maintenance. It will be important in future to resolve how these adhesion machineries cooperate.

## Vinculin links cellular tension to EB differentiation

As vinculin is a well-known mechanosensor, and we showed its function in homeostasis depends on its open active form, we next examined whether vinculin in EBs responds to changes in mechanical tension. To alter the cortical tension of progenitor cells, we modulated myosin II activity. *Drosophila* myosin II is composed of two regulatory light chains and a heavy chain, encoded, respectively, by *spaghetti-squash* (*sqh*), *myosin light chain-cytoplasmic* (*mlc-c*), and *zipper* (*zip*) (*Franke et al., 2006*). The activity of myosin II can be elevated experimentally by expression of a phospho-mimetic, active, form of Sqh (*Mitonaka et al., 2007*). Expression for 14 days resulted in small round progenitor cells (similar to dormant EBs described in *Rojas Villa et al., 2019*), that remained undifferentiated in control guts (*Figure 7A*, *Figure 7—figure supplement 1*). Instead, RNAi-mediated downregulation of *sqh* or *zip* (predicted to reduce cell contractility) mildly increased differentiation rates, and many EBs cells changed shape and were visibly more advanced towards EC differentiation (*Figure 7—figure supplement 1*), reminiscent of the semi-differentiated status of *vincRNAi*-expressing EBs. These results show that altering cell contractility directly impacts progenitor differentiation. We then asked

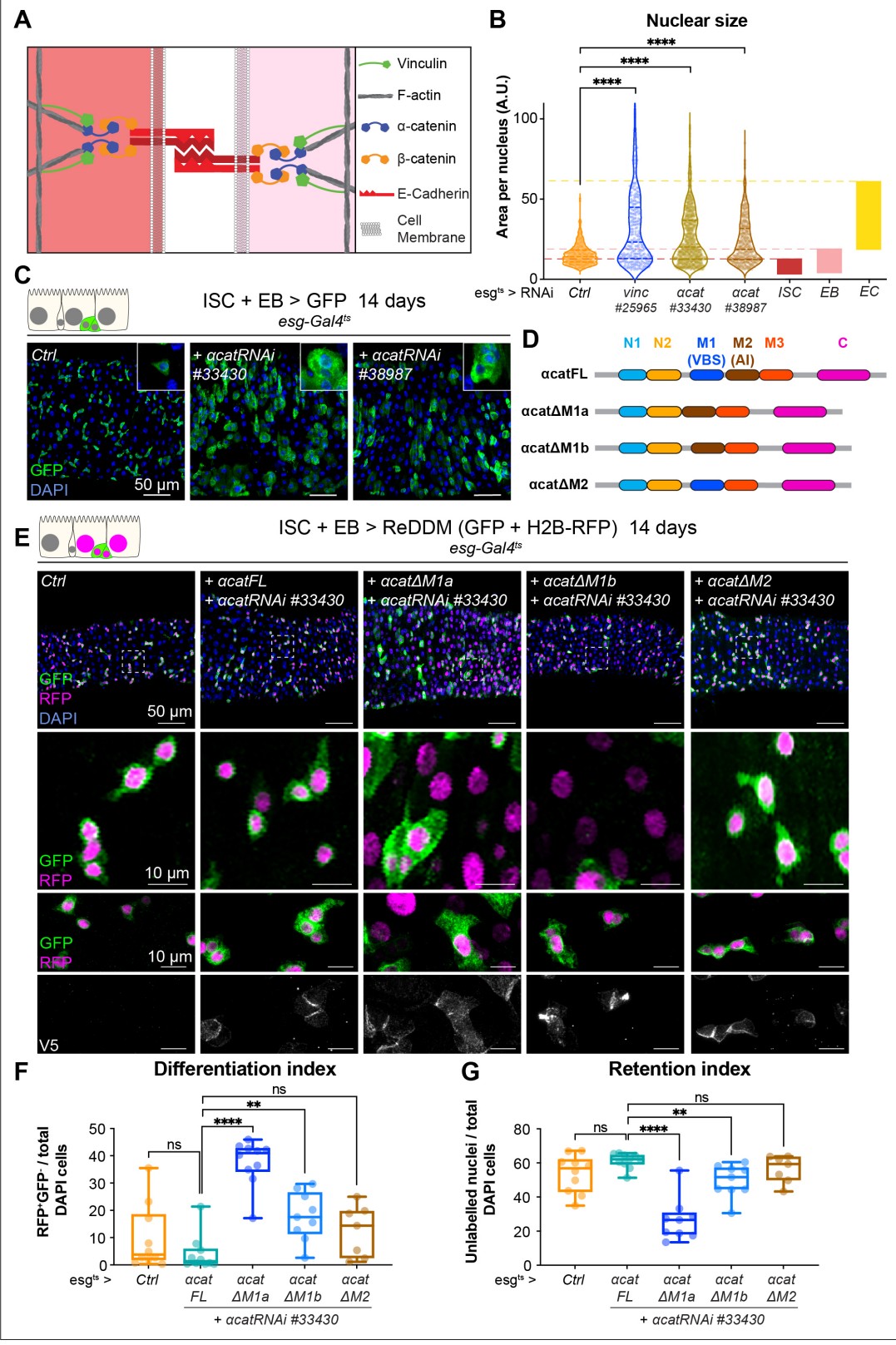

**Figure 6.** Vinculin recruitment to α-catenin prevents enterocyte (EC) differentiation. (**A**) Cartoon depicting activated vinculin binding to α-catenin and actin filaments at adherens junctions. (**B, C**) Expression of two different α-*cat* RNAi in progenitors (marked by green fluorescent protein [GFP], green) for 14 days. (**B**) Quantification of nuclear size in GFP⁺ cells for indicated genotypes. Cells were recorded from a minimum of 3 guts per genotype.

*Figure 6 continued on next page*

*Figure 6 continued*

The coloured boxes intestinal stem cell (ISC), enteroblast (EB), and EC on the right-hand side of the graph indicate nuclear size distributions of each cell type determined in *yw* guts (as in *Figure 4D*). Expression of *vinc* or *α-cat* RNAi produces an accumulation of cells with nuclear size equivalent to large EBs and ECs. (C) Representative images of guts analysed in B. The insets show clusters of GFP cells, comparable to those observed when *vincRNAi* is expressed (see *Figure 1D*). (D) Summary of the rescuing α-catenin constructs used. FL: full-length α-cat; ΔX: full length depleted of indicated domain. The different α-cat domains are coloured. M1 contains the vinculin-binding site (VBS). M2 contains the auto-inhibitory domain. (E) Expression of ReDDM in progenitors with α-*cat* RNAi[#33430] and various V5-tagged α-cat rescuing constructs for 14 days. Ctrl represents expression of ReDDM only. Nuclei are blue (DAPI). Second row shows high magnifications without DAPI staining. Bottom high-magnification panels include V5 staining (white), showing accumulation of α-catenin at cell junctions in all conditions, and cytoplasmic enrichment of the ΔM1a fusion protein. (F) Quantification of the differentiation index (proportion of newly differentiated GFP⁻RFP⁺ cells) indicates that expression of α-cat ΔM1a in α-*cat* RNAi accelerates differentiation. (G) Quantification of the proportion of unlabelled cells indicates the cell retention rate. Expression of α-cat ΔM1a in α-*cat* RNAi accelerates tissue turnover. Two-tailed Mann-Whitney tests were used: ns: not significant, ****$p < 0.0001$, **$p < 0.01$.

The online version of this article includes the following source data and figure supplement(s) for figure 6:

**Source data 1.** Raw data for *Figure 6*, *Figure 6—figure supplement 1*.

**Figure supplement 1.** Vinculin recruitment to enteroblast (EB) cell junctions is reduced in the absence of α-catenin, but α-catenin localises independently of vinculin.

**Figure supplement 2.** Vinculin recruitment to α-catenin in enteroblasts (EBs) but not intestinal stem cells (ISCs) alters tissue turnover.

**Figure supplement 3.** Enteroblast (EB) knockdown of α-catenin and E-cadherin results in EB differentiation.

**Figure supplement 4.** Integrin-mediated adhesion contributes to enteroblast (EB) differentiation.

---

how cellular tension affects vinculin in EBs. Expression of Sqh^DD in *vinc*^{102.1} progenitor cells did not prevent the accelerated differentiation (*Figure 7A, B*), suggesting that vinculin acts downstream of cellular contractility to regulate tissue turnover. As we also observed increased vinculin recruitment to EB adherens junctions when Sqh^DD was expressed in these cells (*Figure 7C, D*), we concluded that vinculin functions downstream of cortical tension, and is mechanosensitive like in other tissues. In addition, we observed that a constitutively open form of vinculin (*Maartens et al., 2016*) slowed down tissue turnover in *vinc*^{102.1} guts (*Figure 7—figure supplement 2*). It is well known that too much vinculin activity (like in this experiment) can be deleterious (*Maartens et al., 2016*), nevertheless, this result is in agreement with a role of vinculin activation in progenitors to prevent premature cell differentiation. Altogether, our data support a model whereby vinculin associates with the VBS of α-catenin in EBs, which becomes exposed upon force-induced α-catenin conformational change. This binding prevents premature EB differentiation and increased ISC proliferation. The idea that vinculin functions by mediating increased tension on adherens junction was supported by the effects of independently altering acto-myosin function.

## Flies with reduced vinculin levels in the intestine are more resilient to starvation

Our experiments have focussed on the gut in a homeostatic state, however plasticity is an important component of intestinal function with stem cell proliferation and progenitor differentiation able to transiently alter in response to diet and regenerate tissue after damage, before reverting to steady-state levels (e.g. *Buchon et al., 2010*; *O'Brien et al., 2011*). Since vinculin has a role in preventing premature EB differentiation and increased ISC proliferation, we hypothesised that *vinc* mutants might display perturbed intestinal regeneration.

To test this, we subjected flies to a cycle of starvation and refeeding as starvation results in cell depletion and gut shortening, whilst refeeding induces cell proliferation and gut growth (*McLeod et al., 2010*; *O'Brien et al., 2011*; *Lucchetta and Ohlstein, 2017*). Cohorts of mated *yw* or *vinc*^{102.1} females were either continuously fed on cornmeal food for the duration of the experiment or fed for 7 days, starved for 7 days (provided with water only) and then refed for a maximum of 14 days. Guts were dissected at various time points (*Figure 8—figure supplement 1A, B*), and gut length and width measured. We observed expected reversible changes in gut width and length (*Figure 8—figure*

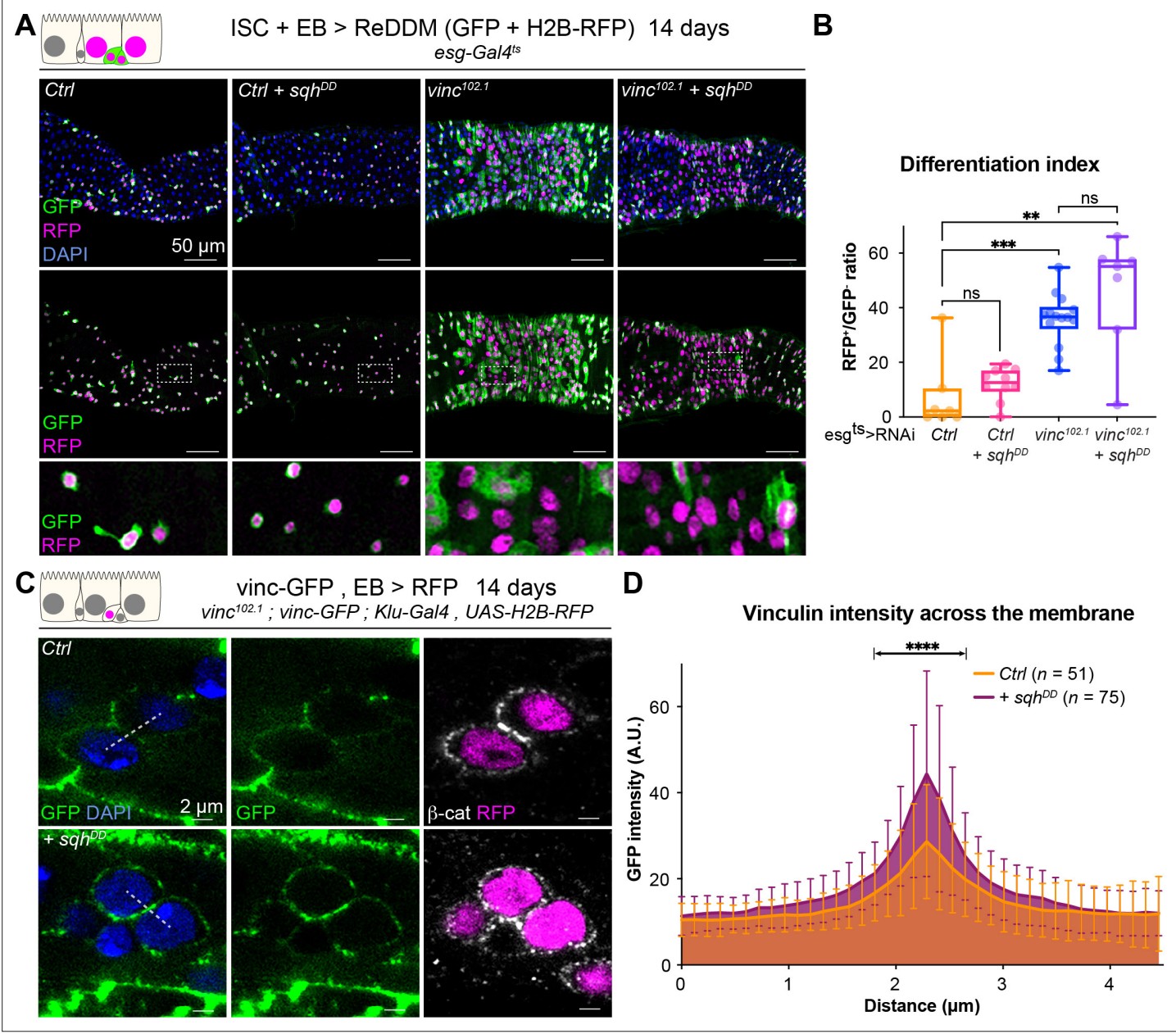

**Figure 7.** Vinculin links cellular tension to enteroblast (EB) differentiation. (**A**) Expression of ReDDM in progenitors with *Sqh^DD* for 14 days, in *yw* (Ctrl) or *vinc^102.1* guts. Top panels show the overlay of RFP, green fluorescent protein (GFP), and DAPI (nuclei, blue). Middle and bottom panels show RFP and GFP only. Bottom panels show high-magnification pictures of the boxes outlined in middle panels. Progenitors round up and remain small upon constitutive activation of Sqh. (**B**) Quantification of the differentiation index (proportion of newly differentiated GFP⁻RFP⁺ cells) indicates that loss of *vinc* expression accelerates differentiation, even upon myosin activation. (**C**) Localisation of Vinculin tagged with GFP (*Vinc-GFP*, green) across cellular junctions (β-cat, white) of progenitor doublets containing at least one EB (labelled by expression of Histone-RFP in *Klu*-expressing cells, red) expressing *Sqh^DD* or not (Ctrl). Dotted lines represent examples of regions used to measure GFP intensity. (**D**) Quantification of Vinc-GFP intensity (arbitrary units, A.U.) from doublets, centred at the highest intensity value (corresponding to the cell/cell junction) indicates increased vinculin recruitment at cell junctions upon expression of *Sqh^DD*. Two-tailed Mann-Whitney tests were used: ns: not significant, ****p<0.0001, ***p<0.001, **p<0.01.

The online version of this article includes the following source data and figure supplement(s) for figure 7:

**Source data 1.** Raw data for *Figure 7*.

**Figure supplement 1.** Myosin activity contributes to progenitor differentiation.

**Figure supplement 2.** Expression of an active form of vinculin slows down tissue turnover in *vinc^102.1* guts.

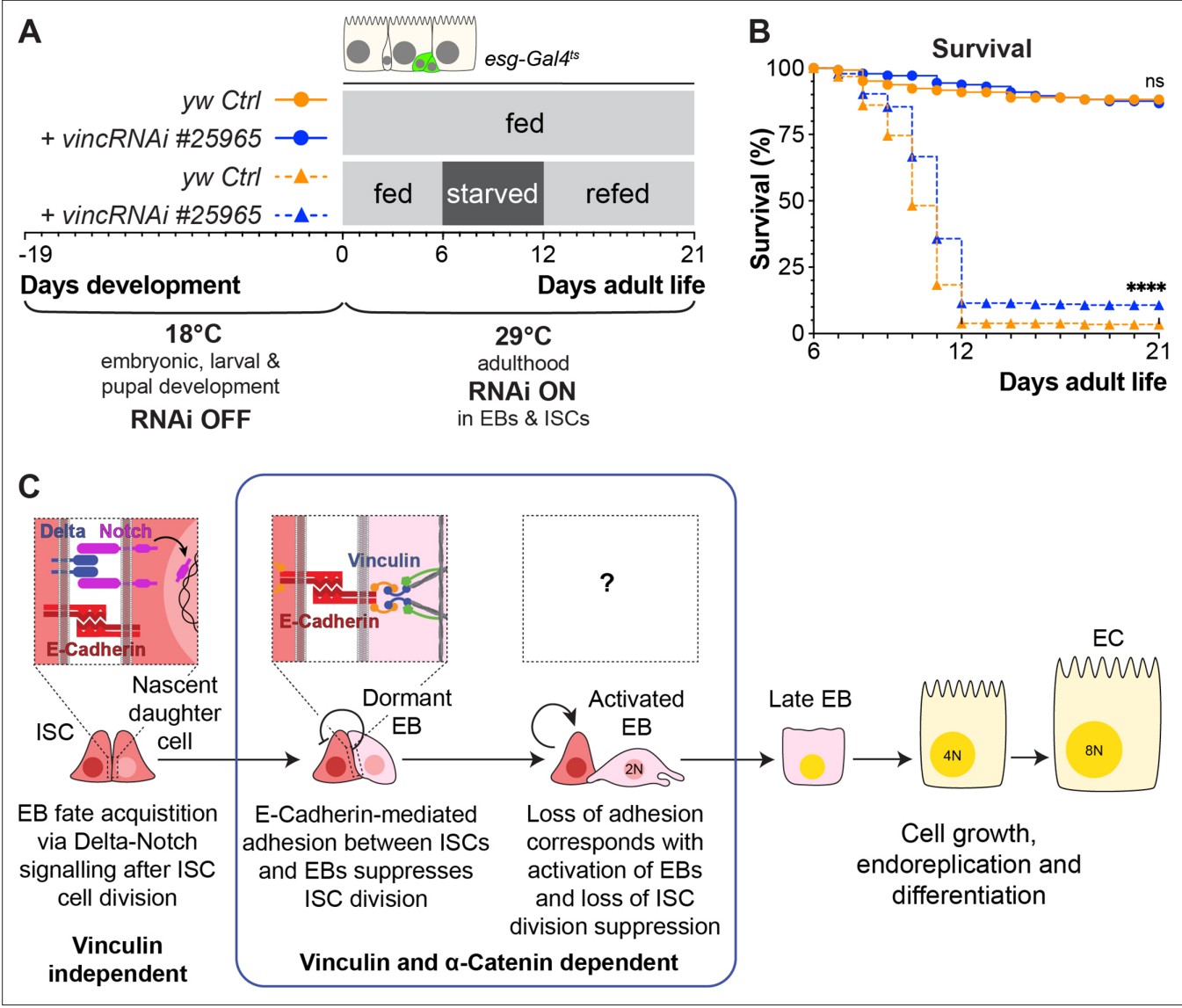

**Figure 8.** Vinculin knockdown in progenitor cells promotes recovery after starvation. (**A**) Schematic of *vincRNAi#25965* expression in intestinal stem cells (ISCs) and enteroblasts (EBs) and feeding regime for flies expressing *vincRNAi#25965* in ISCs and EBs. (**B**) Survival during cycle of feeding, starving and refeeding as shown in (**A**). VincRNAi-expressing flies survive better than control flies. p-values for survival were obtained using a Log-rank test. ns: not significant, ****p<0.0001. (**C**) Model: reinforcement of cell adhesion by α-catenin and vinculin helps to maintain EBs in a dormant state. The mechanical or chemical cues that relieve vinculin from the adhesion complex and promote tissue renewal and enterocyte (EC) differentiation remain to be identified.

The online version of this article includes the following source data and figure supplement(s) for figure 8:

**Source data 1.** Raw data for *Figure 8*.

**Figure supplement 1.** Loss of vinculin activity promotes survival following a period of starvation.

**Figure supplement 1—source data 1.** Raw data for *Figure 8—figure supplement 1*.

supplement 1C, D) for both genotypes, indicating that vinculin is dispensable for diet-induced tissue clearance and repopulation. However, *vinc102.1* flies showed better survival to the starvation regime compared to *yw* flies (*Figure 8—figure supplement 1E*). As this improved resilience to starvation could be a result of vinculin loss in the gut or other tissues, we compared the survival of starved flies when *vincRNAi* was expressed in adult intestinal progenitor cells. Again, we observed better survival in the absence of vinculin (*Figure 8A, B*). We propose that the larger number of progenitor cells resulting from vinculin depletion could have minimised the deleterious effects of cell loss and/or provided a greater pool of cells ready to start differentiating in the right nutritional conditions. Whilst

we cannot exclude that vinculin may play important roles in other tissues to regulate organismal physiology throughout development and adulthood, our result shows an important contribution of vinculin in progenitor cells to adult fly survival.

These experiments lead us to propose a model whereby vinculin functions within EBs to maintain their dormancy and prevent adjacent ISCs from proliferating, most likely by increasing the force on cell junctions (*Figure 8C*). The mechanical or chemical cues that relieve vinculin from the adhesion complex and promote tissue renewal and EC differentiation remain to be identified.

## Discussion

Recent *in vitro* manipulation of tissues, cells, and substrates have demonstrated the importance of mechanotransduction in cell fate specification (*Kumar et al., 2017*). Here, we employed genetic approaches to establish *in vivo* the role of the conserved mechanoeffector protein vinculin in intestinal tissue homeostasis.

### Vinculin-dependent junctional stability regulates intestinal tissue turnover

Using the *Drosophila* intestine as a simple model of an established self-renewing adult epithelium, we have discovered that vinculin regulates EB differentiation. With conditional and cell-specific knockdowns, we demonstrated that unlike two core members of cell–matrix adhesion complex, the integrin βPS subunit and talin, which directly binds vinculin, vinculin appears dispensable for ISC maintenance. This was surprising to us given the well-known role of vinculin in focal adhesion stabilisation *in vitro* (reviewed in *Carisey and Ballestrem, 2011*; *Bays and DeMali, 2017*). Our work instead showed that vinculin depletion in EBs induces their partial maturation towards absorptive cells, in turn accelerating tissue turnover. As this function is dependent on interaction with α-catenin, we concluded that vinculin may be required in EBs to stabilise adherens junctions. Interestingly, altering E-cadherin- or integrin-mediated adhesion in EBs prevented premature differentiation. Since vinculin can modulate both adhesion machineries, the crosstalk between them remains to be characterised in this tissue. As the *vinc* phenotype correlated with that caused by decreased levels of myosin II activity it is likely that in the intestine vinculin senses myosin II tension, as in other developing epithelia in mammals (*Rübsam et al., 2017*) and *Drosophila* (*Case et al., 2015*; *Jurado et al., 2016*; *Kale et al., 2018*; *Alégot et al., 2019*). This could explain why in ISC/EB pairs, vinculin is only required on the EB side where cytoskeletal tension vary as cells undergo morphological changes during maturation and differentiation. Recent work in cell culture models showed that vinculin can be recruited asymmetrically in a force-dependent manner at sites of cadherin adhesion near mitotic cells. Whilst in this case, asymmetric recruitment promotes cell shape changes during cell division and maintenance of epithelial integrity (*Monster et al., 2021*), our study reveals that asymmetric vinculin requirements at cell junctions (which we showed is influenced by myosin II activity) can also regulate cell fate decisions and tissue composition in the intestine. Our findings place vinculin in a feedback loop that couples the process of EC differentiation to stem cell proliferation in the intestine. In the absence of vinculin, ISCs constitutively sense a demand for tissue replenishment and produce new cells, despite the tissue containing high proportions of progenitors, thus generating increased cell density. Whilst our data mostly support a direct role of vinculin at cell junctions in EBs, we cannot exclude that vinculin activity might also impact long range signals regulating stem cell proliferation, especially in light of the mild effects observed when vinculin is depleted in small clones of cells rather than globally (*Figure 4E–G*).

Previous work showed that in response to poor nutritional conditions (or insulin signalling pathway impairment), adherens junctions between ISCs and newly formed EBs were stabilised and stem cell proliferation was prevented (*Choi et al., 2011*). Although the signals linking insulin signalling to junction disassembly and non-autonomous proliferation remain to be characterised, it is tempting to speculate that α-catenin recruitment of active vinculin might help strengthen adherens junctions when cells are under external tension, as would be the case during periods of active growth. In support of this hypothesis, upon refeeding (and thus active insulin signalling) after a week of starvation, $vinc^{102.1}$ guts showed a recovery in size that initially surpassed the original one, correlating with increased cell production facilitated by junction disassembly. It will be interesting in future to establish whether in this system vinculin acts in parallel or downstream of the insulin signalling pathway.

Notably, E-cadherin-mediated cell adhesion is also required in ECs to prevent stem cell renewal, but in this case, by preventing the secretion of mitogenic signals (*Liang et al., 2017*). Such coupling of differentiated cell density and self-renewal was also demonstrated *in vivo* in the mouse adult stratified epidermis, where stem cells divide in response to delamination (*Mesa et al., 2018*).

## Mechanosensing in intestinal precursors regulates tissue homeostasis

It seems fitting that proteins activated in response to tissue tension couple differentiation to self-renewal. Strikingly, we observed regional differences in the rate of tissue turnover in the posterior midgut of $vinc^{102.1}$ guts. As the region with highest turnover in the mutant corresponded to a narrowing portion of the gut where the domains R4 and R5 are separated by a constriction of the intestinal wall (cartoon *Figure 1B*, *Buchon et al., 2013*; *Marianes and Spradling, 2013*), we speculate that a local increase in mechanical forces (e.g. through visceral muscle contractions and food passage) induces vinculin mediated junction reinforcement to prevent untimely cell production. Consistent with this, β-catenin had a stronger cortical localisation in this area compared to more anterior areas of the midgut (data not shown). Another hypothesis is that this gut area presents with differential expression of signalling components. These issues remain to be solved.

Our work demonstrated that vinculin is required in EBs to prevent their precocious activation and differentiation into absorptive cells. How vinculin relates to transcription factors activated at this stage such as Esg and Klu remains to be formally tested. However, we observed that *vinc* knockdown in EBs cells led them to stall half-way through differentiation, whilst triggering proliferation in adjacent ISCs. This suggests that vinculin prevents the premature activation of dormant EBs. Once past a certain threshold of mechanical force, vinculin might not be sufficient to prevent tissue growth. Alternatively, vinculin might also contribute to terminal differentiation, either by direct signalling of terminal differentiation or by sensing local tissue density in a similar fashion as contact inhibition (*Alégot et al., 2019*; *Biswas et al., 2021*). Indeed, gut stretching after food ingestion induces proliferation through activation of the YAP/TAZ homolog Yorkie. The mechanical stress induces a displacement of the Ste20 kinase Misshapen from the membrane of EBs, thereby relieving interactions with upstream mediators of the Hippo pathway, and thus allowing Yorkie nuclear translocation, activation of target genes and midgut growth (*Li et al., 2018*). We suggest that vinculin contributes to mechanical force sensing in EBs and more generally helps coordinate self-renewal and differentiation into absorptive cells. Interestingly, the precursors of EEs can also transduce mechanical stress to control differentiation. In this case, however, cell fate is acquired through increased intracellular calcium downstream of the stretch-activated channel piezo expressed in EE precursors (*He et al., 2018*).

Thus, lineage commitment in the *Drosophila* intestine seems to depend on mechanosensing pathways specifically active in precursors cells. Together with the role of integrin activation in ISCs to regulate self-renewal (this work; *Lin et al., 2013*), it becomes apparent cell type-specific mechanosensing pathways contribute to fine-tuning cell production and tissue composition.

## Outlook

In the mammalian intestine, stem cells are located in crypts whilst progenitors concentrate at the bottom of villi, which are populated by differentiated cells. Epithelial turnover is dependent on E-cadherin mediated adhesion (*Hermiston and Gordon, 1995*; *Hermiston et al., 1996*) and continuous cell migration (*Krndija et al., 2019*) thus, it is harder to establish if forces contribute to proliferation and lineage commitment or active cell migration (and indirectly self-renewal). The mechanoeffector protein vinculin is deregulated in gastrointestinal diseases, with expression levels reduced in colorectal cancers (*Goldmann et al., 2013*; *Li et al., 2017*) and auto anti-vinculin antibodies are produced in inflammatory bowel syndrome (*Rezaie et al., 2017*). The use of a simpler model organism, with a flat intestinal epithelium where stem cells are dispersed among their progeny (*Micchelli and Perrimon, 2006*; *Ohlstein and Spradling, 2006*) and cell migration is sporadic (*Antonello et al., 2015*; *Martin et al., 2018*) has allowed us to identify an EB-specific role for vinculin mediated by α-catenin and independent of integrin adhesion. It will be important in future to develop tools to experimentally define how and when vinculin is activated to strengthen cell junctions in the healthy intestine and how to best harness these data with regard to vinculin deregulation in disease.

## Materials and methods

### Fly strains and experimental conditions

All *Drosophila* stocks were maintained at 18°C or amplified at 25°C on standard medium (cornmeal, yeast, glucose, agar, water, Nipagin food medium). The following strains were used: *yw*, *Vinc-GFP* (*Klapholz et al., 2015*), *Vinc-RFP* (*Klapholz et al., 2015*), *vinc$^{102.1}$* (*Klapholz et al., 2015*; a deletion that removes *vinc*, *SmydA-8* and partially *Mct1*), UAS-Vinc$^{CO}$-RFP (*Maartens et al., 2016*), UAS-sqh$^{DD}$, UAS-sqhRNAi, UAS-zipperRNAi (gifts from Silvia Aldaz), *esg$^{ts}$-FlipOUT* (*Jiang et al., 2009*), *esg$^{ts}$Gal4* (expressed in progenitor cells, *Jiang et al., 2009*), esg-Gal4$^{ts}$Su(H)Gal80 (expressed in *Wang et al., 2014*), esg$^{ts}$-REDDM (*Antonello et al., 2015*), Klu$^{ts}$-REDDM (expressed in EBs, *Antonello et al., 2015*), *Klu$^{ts}$Gal4* (gift from Tobias Reiff), MyoIA$^{ts}$Gal4 (expressed in *Jiang et al., 2009*), *vincRNAi* #25965 (Bloomington, TRiP.JF01985), *vincRNAi* #41959 (Bloomington, TRiP.HMS02356), NRE-lacZ (*Furriols and Bray, 2001*), 10XSTAT-GFP (*Bach et al., 2007*), mys RNAi #27735 (Bloomington, TRIP.JF02819), talin RNAi #28950 (Bloomington, TRIP.HM05161), talin RNAi #33913 (Bloomington, TRIP.HMS00856), α-Cat RNAi #33430 (Bloomington, TRIP.HMS00317), α-Cat RNAi #38987 (Bloomington, TRIP.HMS00837), and shg RNAi #32904 (Bloomington, TRiP.HMS00693). The following lines were gifted by K. Irvine (*Alégot et al., 2019*): α-Cat-FL-V5; α-Cat RNAi #33430, α-CatΔM1a-v5; α-Cat RNAi #33430, α-CatΔM1b-v5; α-Cat RNAi #33430, α-CatΔM2-v5; α-Cat RNAi #33430 – see Supplemental material for complete genotypes.

### RNAi knockdown

RNAi knockdown efficiencies were tested by immunostaining (*mys*, *talin*, and α-*catenin*) or by RT-qPCR (*vincRNAi*s). Transcripts were downregulated by UAS-driven RNAi conditional expression in various cell populations using appropriate Gal4 driver lines combined with the thermosensitive repressor Gal80, to restrict expression to just adult stages, by shifting to 29°C, the permissive temperature for Gal4 expression as its thermosensitive repressor Gal80 stops functioning. To prevent RNAi expression during development, flies were kept at restrictive temperature (18°C) until 3 days after hatching, after which 10–15 mated females were transferred at the permissive temperature for a minimum of 14 days to allow intestinal expression of UAS-driven constructs. Food was changed every 2 days throughout the experiment.

### Escargot-Flip-Out experiments

*flippase* expression under the control of *esg-Gal4* in progenitor cells, at the permissive temperature for Gal4 expression (29°C), excises a transcriptional stop between a ubiquitous *actin* promotor and Gal4, resulting in permanent, heritable expression of Gal4 in all progenitor cells and their progeny, as revealed with UAS-GFP (*Jiang et al., 2009*, *Figure 2C–G*, and *Figure 2—figure supplement 2A*). Flies were kept at restrictive temperature for Gal4 expression (18°C) until 3 days after hatching, after which 10–15 males were transferred at the permissive temperature for a minimum of 14 days. Food was changed every 2 days throughout the experiment.

### ReDDM experiments

The lineage tracer ReDDM (Repressible Dual Differential stability cell Marker) is a UAS-driven transgene encoding a mCD8-GFP with a short half-life, and H2B-RFP with a long half-life both contained in two independent UAS-constructs (*Figure 4* and *Figure 5*; *Antonello et al., 2015*). When expressed under the control of *esg-Gal4* or *Klu-Gal4*, this system allowed us to identify unambiguously newly differentiated ECs as RFP$^+$GFP$^-$ cells and thus establish the rate of differentiation.

### Dissection and immunostaining

Adult guts were dissected in phosphate-buffered saline (PBS) and fixed twice during 20 min in fresh 4% paraformaldehyde diluted in PBS (18814-10, Polysciences) with a 10 min wash in PBT 0.1% (PBS + Triton X-100, Sigma-Aldrich) in between, apart for *Figure 2B–F*, where guts were fixed once. Samples were then washed 5 min three times in PBT 0.1% and permeabilised for 30 min in PBT 1%. All samples were then incubated 30 min at room temperature in blocking buffer (PBS 0.1% + 1% bovine serum albumin, A2153-10G, Sigma) followed with primary antibodies incubation in blocking buffer overnight at 4°C. Samples were then washed three times 15 min in PBT 0.1%, subjected to secondary

antibody staining in blocking buffer for 2 hr at room temperature followed by three washings in PBT 0.1%. Guts were mounted in Vectashield (H-100, Vector Laboratories) between coverslip and glass slide. The following primary antibodies were used: chicken anti-GFP, 1/1000 (Abcam 13970); mouse anti-Delta, 1/1000 (C594.9B-c, DSHB); chicken anti-Beta galactosidase, 1/1000 (Abcam 9361); mouse anti-Armadillo (β-catenin), 1/50 (N2 7A1-c, DSHB), mouse anti-V5, 1/400 (R96025, Thermo Fisher), rat anti-α-catenin, 1/20 (DSHB D-CAT1), rabbit anti-PH3(Ser10), 1/500 (9701 CST); mouse anti-Prospero, 1/50 (DSHB, MR1a). Alexa-488-, 555-, and 647-conjugated secondary goat antibodies or Phalloidin were used (Molecular probes) and nuclei were counterstained with DAPI.

## Imaging and analysis

All confocal images were taken on a Leica SP8 Confocal microscope with ×20 air lens, ×40 and ×63 oil immersion lenses. 405, 488, 516, and 633 nm lasers were used as appropriate, with gain and power consistent within experiments. For z-stacks, images were acquired every 1 µm through the tissue. Whole length guts shown in *Figure 1* and *Figure 2* are tile scans acquired directly on the Leica SP8. Brightfield whole gut photographs were taken on a Zeiss Stemi 305 connected to an iPad or a Leica M165FC microscope with Leica DFC3000G camera. All images were analysed with FiJi and Adobe Photoshop CS. When necessary, images were stitched using the ImageJ pairwise stitching plugin. Scale bars are indicated within panels.

## Feeding, starvation, refeeding regime, and survival

Crosses of *esgGal4; UAS GFP; tubGal80$^{ts}$* virgins and *yw* or *vincRNAi$^{#25965}$* males were set up in bottles of propionic food (to improve laying), placed at 18°C and flipped every 3 days. After 19 days at 18°C, bottles were transferred to 29°C for 3 days to allow flies to hatch. Hatchlings were collected, transferred to fresh bottles of standard medium, and kept at 29°C for 3 days to allow flies to reach sexual maturity and mate. Flies were then anesthetised for minimal time to retrieve female flies, which were transferred into vials of either standard medium or starvation conditions – vials containing a cellulose acetate bung (Fly1192, Scientific Laboratory Supplies) soaked in water – at a density of 12 flies per vial. All vials were placed at placed horizontally at 29°C, with vial position randomised to control for any variation in temperature or humidity in the incubator. Survival was recorded daily. Control flies were transferred to fresh food every 2 days and starvation flies had the bung replaced every 24 hr to prevent drying. After 6 days of starvation, starved flied were placed back into vials of standard medium. All flies were transferred to fresh food every 2 days, and survival continued to be recorded daily for the remainder of the experiment. Significance was determined using the log-rank test.

For analysis of *yw* and *vinc$^{102.1}$* flies, stocks were reared in parallel in bottles of standard medium at room temperature for several generations prior to the experiment. Flies used in the experiment underwent embryonic, larval, and pupal development in bottles of standard medium at room temperature, except for the final 2 days before hatching when bottles were placed at 25°C to speed up development. After discarding early emergents, bottles were placed back at 25°C for 2 days, after which all synchronised hatchlings were collected and transferred to new bottles of standard medium. These bottles were placed at 25°C under a 12-hr light–dark cycle for 2 days to allow flies to reach sexual maturity and mate. Flies were then anesthetised for minimal time to retrieve female flies, which were transferred into vials of standard medium at a density of 12 flies per vial. Vials were placed horizontally at 25°C under a 12-hr light–dark cycle, with vial position randomised to control for any variation in temperature or humidity in the incubator. Survival was scored each morning, and flies were transferred to new vials containing fresh food every 2 days. When flies were 7 days old, 12 flies per genotype were dissected in PBS (BR0014G, Oxoid) and imaged immediately to measure midgut length (between the centre of the proventriculus and the Malpighian tubule attachment site, between arrows in *Figure 6A*) and width in the R4/5 region. Of the remaining flies for each genotype, approximately half were transferred to starvation conditions – vials containing a cellulose acetate bung (Fly1192, Scientific Laboratory Supplies) soaked in mineral water (Shropshire Hills Mineral Water, Wenlock Spring Water Limited). The bung was replaced every 24 hr to prevent drying. Control flies were continuously fed in vials of standard medium. After 7 days of starvation, 12 flies from each genotype and condition ('fed-starved' and 'continuously fed') were dissected and midgut length and width measured. Remaining 'fed-starved' flies were then transferred back into vials of standard medium. After 4 days of refeeding, 12 flies from each genotype and condition ('fed-starved-refed' and 'continuously fed') were dissected

and midgut length and width measured. All remaining flies continued to be reared in vials of standard medium with dissections performed every 3–4 days, until the flies reached 28 days old at which point the experiment ended. Throughout the starvation and refeeding assay, survival was scored each morning. Any flies which left the experiment prior to natural death were censored and removed from the survival analysis.

## Quantifications and statistics

Unless otherwise indicated the images shown in this study and the respective quantifications focused on the specific R4/5 region of the posterior midgut. *ISC*, *EB*, and *progenitor ratios* were quantified manually through the cell counter plugin in ImageJ software and represented as the ratio of GFP$^+$ cells over the total number of nuclei in DAPI. *Cell density* was assessed semi automatically through binarisation of DAPI signal in ImageJ and measurement of the number of nuclei in a fixed area (10,000 μm$^2$) representative of the whole images processed. The *differentiation index* was assessed by manually counting GFP$^+$RFP$^+$ and GFP$^-$RFP$^+$ cells via the cell counter plugin in ImageJ. The differentiation index was calculated as the ratio of differentiated cells (GFP$^-$RFP$^+$) number over total cell numbers (DAPI$^+$). *Nuclear size* was measured by binarising the DAPI signal of all nuclei using ImageJ. Cell type was defined using anti-Dl (ISC) antibody staining, Su(H)-lacZ (EB) reporter. Small DAPI$^+$Dl$^+$ or DAPI$^+$LacZ$^+$ cells (mutually exclusive stainings) located near the basal side of the epithelium were quantified manually on z-stack reconstructions spanning half the depth of the intestine of samples. To exclude EEs from EC nuclear size measurement only Dl$^-$, Su(H)$^-$, and bigger than ISC nuclear size were considered. For cross-sectional nuclei counting, confocal z-stacks through the full midgut depth were visualised with the ImageJ orthogonal view tool. The number of epithelial cell nuclei and maximum midgut width was measured in progressive cross-sections every 30 μm along 300 μm of the R4/5 region for both genotypes. *GFP intensity* reporting vinc-GFP recruitment at cell junctions (determined by β-catenin/Arm staining) was measured from a single z-section further away from the visceral muscles to avoid any confounding GFP signal. On Fiji, a straight line (width = 9) was drawn in between the centre of ISC/EB (Klu$^-$/Klu$^+$) and EB/EB (Klu$^+$/Klu$^+$) doublets nuclei. The plot profile option was then used to quantify GFP intensity across the previously drawn line. After extracting the numerical data, quantifications from each doublet were then centred at the highest intensity value (corresponding to the cell/cell junction). Data were plotted with GraphPad Prism 7, 8, and 9 software. In all graphs, results represent median values, error bars represent interquartile range. Statistical significance was calculated using a Mann–Whitney test and p values <0.05 were considered statistically significant. Survival was compared between *yw* and *vinc*[102.1] flies using the log-rank test.

## Wing mounting

Wings were mounted in Euparal (R1344A, Agar Scientific) on a glass slide (SuperFrost 1.0 mm, ISO 8037/1, VWR International) with coverslip (22 × 50 mm #1, 12342118, Menzel-Gläser). Images were taken on a Zeiss Axiophot microscope with a Q Imaging QICAM FAST1394 camera.

## Scanning electron microscopy

Guts of 7-day-old female flies were dissected in Schneider's medium (S3652, Sigma) and fixed overnight at 4°C in 2% glutaraldehyde, 2% formaldehyde in 0.05 M sodium cacodylate buffer (pH 7.4) containing 2 mM CaCl$_2$. After fixation, guts were cut at the middle point of the posterior midgut with a razor blade (WS1010, Wilkinson Sword). Guts were washed with 0.05 M sodium cacodylate buffer and osmicated overnight at 4°C in 1% OsO$_4$, 1.5% potassium ferricyanide, and 0.05 M sodium cacodylate buffer (pH 7.4). Guts were washed in deionised water (DIW) and treated for 20 min in the dark at room temperature with 0.1% thiocarbohydrazide/DIW. Guts were washed again in DIW and osmicated for 1 hr at room temperature in 2% OsO$_4$ in DIW. Following washing in DIW, guts were treated for 3 days at 4°C in 2% uranyl acetate in 0.05 M maleate buffer (pH 5.5). Guts were washed in DIW and stained for 1 hr at 60°C in lead aspartate solution (0.33 g lead nitrate in 50 ml 0.03 M aspartic acid solution [pH 5.5]). Following washing in DIW, guts were dehydrated in 50%, 70%, 95–100% ethanol, three times in each for at least 5 min each. Guts were then dehydrated twice in 100% dry ethanol, twice in 100% dry acetone, and three times in dry acetonitrile for at least 5 min each. Guts were placed for 2 hr at room temperature in a 50/50 mixture of Quetol (TAAB) resin mix and 100% dry acetonitrile. Guts were incubated in pure Quetol resin mix (12 g Quetol 651, 15.7 g NSA, 5.7 g MNA, and 0.5 g BDMA [all from

TAAB]) for 5 days, replacing with fresh resin mix each day. Guts were embedded in coffin mounds and incubated for 48 hr at 60°C. Resin-embedded samples were mounted on aluminium SEM stubs using conductive epoxy resin and sputter coated with 35 nm gold. Blockfaces were sectioned using a Leica Ultracut E and coated with 30 nm carbon for conductivity. The samples were imaged in a FEI Verios 460 SEM at an accelerating voltage of 3–4 keV and a probe current of 0.2 pA in backscatter mode using the CBS detector in immersion mode. Large area maps were acquired using FEI MAPS software for automated image acquisition. MAPS settings for high-resolution maps were 1536 × 1024 pixel resolution, 10-µs dwell time, two-line integrations, magnification ~×8000, working distance ~4 mm, tile size 15.9 µm, default stitching profiles.

## Fluorescence heatmaps

Posterior midguts were imaged with a Leica M165FC microscope with Leica DFC3000G camera. Fluorescence intensity was measured along three lines of defined position ('upper quarter', 'middle', and 'lower quarter') spanning the length of the posterior-most 1200 µm of each midgut using the ImageJ plot profile tool. Fluorescence intensity along the three lines was averaged for each gut. Then average fluorescence intensity was calculated for each 50 µm gut segment and normalised by dividing by the lowest fluorescence value (to account for variation in image brightness within and between genotypes). This was repeated for tens of guts for each genotype, and the overall average normalised values for each 50 µm gut segment were plotted as a heatmap using Prism.

## Acknowledgements

We thank Nick Brown, Sarah Bray, Tobias Reiff, Maria Dominguez, Cedric Polesello, and Ken Irvine for kindly sharing flies with us. *Drosophila* stocks used in this study were otherwise obtained from the Bloomington *Drosophila* Stock Center (NIH P40OD018537). Several antibodies were obtained from the Developmental Studies Hybridoma Bank, created by the NICHD of the NIH and maintained at The University of Iowa, Department of Biology, Iowa City, IA 52242. We thank the Cambridge Advanced Imaging Centre for their help with electron microscopy and for use of the confocal facility. We are grateful to members of the Brown lab (PDN, Cambridge) for insightful suggestions and comments on the project and the manuscript, and Aki Stubb (PDN, Cambridge) for constructive feedback. This work was supported by a Sir Henry Dale Fellowship jointly funded by the Wellcome Trust and the Royal Society to GK [grant number 206208/Z/17/Z] and a Wellcome Trust PhD studentship to BLET [programme number 102175/B/13/Z].

## Additional information

### Funding

| Funder | Grant reference number | Author |
| --- | --- | --- |
| Wellcome Trust | Sir Henry Dale Fellowship (206208/Z/17/Z) | Golnar Kolahgar |
| Wellcome Trust | PhD student studentship (102175/B/13/Z) | Buffy L Eldridge-Thomas |
| Wellcome Trust | Sir Henry Dale Fellowship to GK(206208/Z/17/Z) | Jerome Bohere |

The funders had no role in study design, data collection, and interpretation, or the decision to submit the work for publication. For the purpose of Open Access, the authors have applied a CC BY public copyright license to any Author Accepted Manuscript version arising from this submission.

### Author contributions

Jerome Bohere, Buffy L Eldridge-Thomas, Conceptualization, Data curation, Formal analysis, Validation, Investigation, Visualization, Methodology, Writing - original draft, Project administration, Writing - review and editing; Golnar Kolahgar, Conceptualization, Formal analysis, Supervision, Funding

acquisition, Validation, Investigation, Visualization, Methodology, Writing - original draft, Project administration, Writing - review and editing

### Author ORCIDs
Jerome Bohere http://orcid.org/0000-0001-8305-129X
Buffy L Eldridge-Thomas http://orcid.org/0000-0003-4070-2827
Golnar Kolahgar http://orcid.org/0000-0003-4007-3311

### Decision letter and Author response
Decision letter https://doi.org/10.7554/eLife.72836.sa1
Author response https://doi.org/10.7554/eLife.72836.sa2

## Additional files

### Supplementary files
• Transparent reporting form
• Supplementary file 1. List of experimental genotypes used throughout the figures.

### Data availability
All data generated or analysed during this study are included in the manuscript.

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
