## [Editor Report]

This study from Kolahgar and colleagues examines a potential mechanosensation mechanism in fly intestinal stem cells and their enteroblast progeny. The authors focus on vinculin, whose activity in other epithelial systems is regulated by mechanical tension. The manuscript's data clearly demonstrate a role for vinculin in suppressing the proliferation of midgut stem cells and the differentiation of their enteroblast progeny and suggest that this role is exerted specifically through enteroblast vinculin. The authors find that similar phenotypes are induced by genetic manipulations of vinculin, α-catenin, and myosin, and they argue that this similarity implies that vinculin activity in enteroblasts is mechanosensitive. These findings are potentially relevant to biologists interested in stem cells, tissue homeostasis, fate decisions, and mechanobiology.

---

## [Decision Letter]

**Decision letter after peer review:**

Thank you for submitting your article "Vinculin recruitment to α-catenin halts the differentiation and maturation of enterocyte progenitors to maintain homeostasis of the *Drosophila* intestine." for consideration by *eLife*. Your article has been reviewed by 3 peer reviewers, and the evaluation has been overseen by Lucy O'Brien as Reviewing Editor and Mone Zaidi as the Senior Editor. The following individual involved in review of your submission has agreed to reveal their identity: Joaquin de Navascues (Reviewer #1).

Essential revisions:

(1) Explicitly examine whether vinculin in enteroblasts is regulated by changes in mechanical tension and/or cadherin-mediated adhesion.

A pillar of the authors' model is that enteroblast vinculin exerts its effects through a mechanosensitive mechanism involving E-cad and a-cat. Work in other epithelial tissues has demonstrated that the E-cad-a-cat-vinc complex is mechanosensitive, but this mechanosensitivity has not been demonstrated in the fly gut. Since some canonical functions of E-cadherin are known to differ in the fly gut, it is important to establish that mechanosensitive regulation of vinc is conserved.

The chief evidence the authors present for mechanosensitivity is that single genetic perturbations of vinc and myosin produce similar phenotypes. However, the phenotypic readouts that are shown (nuclear size, stem cell production, differentiation index, retention index) are indicators of stem cell renewal/differentiation. These types of effects could in principle arise through numerous mechanisms, some of which are not necessarily related to tension.

Reviewer 3 suggests several approaches to more directly examine whether vinc in enteroblasts is responding to (or perhaps regulating) changes in mechanical tension. These include epistasis tests with vinc and loss- or gain-of function alleles of myosin, epistasis tests with vinc and E-cad (or perhaps E-cad mutants that lack the extracellular binding domain or the a-cat binding site), or examining the subcellular localization of vinc following loss- or gain-of-function myosin manipulations and/or in a-cat mutants. One of these suggestions, or an equivalent experiment, will be essential in a revised manuscript.

(2) Provide textual explanations and/or experimental evidence to clarify certain cell-specific phenotypes.

All reviewers felt that interpretations of some cell type-specific genetic manipulations were either oversimplified or, potentially, contradictory:

a. There was concern that some experimental outcomes (MARCM expts, knockdown experiments using ISC+EB vs EB drivers) do not support the authors' conclusion that vinc acts in enteroblasts and not in stem cells. Can the authors clarify these apparent discrepancies through either textual discussion or new data?

b. The conclusion that progenitor cell vinc does not act through integrins, which is presented as a key finding, is based on the result that mys or talin RNAi does not phenocopy vinc RNAi. However, these data do not exclude a scenario in which vinc acts through both cadherin and integrins, with the cadherin pathway dominating. Either additional data should be provided that directly test the involvement of integrins, or the text should be modified to discuss the limitations of the current approach.

c. Why does vinc::RFP rescue the vinc null while vincCO has little effect?

(3) As shown, the starvation/refeeding and survival assays are difficult to interpret because their use of whole-animal vinc nulls means that the outcomes cannot necessarily be ascribed to enteroblast vinculin. These experiments should either be repeated in a manner that reveals the contribution of enteroblast vinculin, or else removed from the study (as suggested by Reviewer 1).

(4) Dl immunostaining – Can the authors comment on how quantitation of Dl+ cells was performed with the punctate immunostaining pattern evident in, for instance, Figure 4a-b? In online discussion among the reviewers, there were differing opinions about Dl staining patterns but agreement that ascribing Dl puncta to individual cells may be unreliable without a membrane marker.

*Reviewer #1 (Recommendations for the authors):*

I am mostly repeating myself here but the experiments I would suggest are:

– Repeating some key experiments while recording additional, more standardised metrics, in particular evaluating PH3 and rates of differentiation with ISC-Gal4, esg-Gal4, klu-Gal4 and MARCM clones with UAS-vincRNAi.

– Establishing the autonomy of the phenotype in vinc[102.1] in one of the ways suggested above.

– Establishing the autonomy of the α-catenin phenotype in ISCs and EBs, and whether the full deletion of the M1 region fails to rescue the phenotype.

– Establish whether expression of VincCO (or VincFL) can change the behaviour of the system when forced to regenerate.

Some technical fastidious stuff, sorry:

– Lifespan experiment – individuals dissected for gut staining cannot be assumed to experience the same risk as the rest – the correct approach is to remove them from the cohort or censor them, otherwise the amount of animals at risk after dissection is not the right one.

– Some language mishaps:

There is a couple of instances where esgTS-FO patches are referred to as "clones", it should be corrected.

The authors refer to 29ºC in experiments using tub-Gal80ts as the 'permissive' temperature, which is not the usual way to refer to this in the literature, as this temperature is when Gal80ts stops functioning – I understand that for the activity of Gal4 this is 'permissive' but the primary molecular event is the destabilisation of Gal80ts and that is how the community is used to think of this – if the authors wish to maintain this nomenclature, perhaps some more explicit mention of what they mean would be useful in the main text.

The use of the term "compensatory proliferation" is a bit out of context here, as it has been employed traditionally to refer to proliferation to compensate apoptosis; however, in this paper it is argued that the proliferation arises from loss of contact inhibition due to differentiation, so the phenomenology does not parallel the traditional use of the term in *Drosophila*.

The authors correctly conclude that flies are more resistant to starvation and they word it very similarly in the main text, but in the abstract they phrase this observation as "Improved recovery after starvation", which is misleading as the phenotype is circumscribed to the starvation period: the survival curve is essentially the same AFTER the re-feeding, and the only difference in the re-feeding period is this single time point when the guts from vinc[102.1] guts recovered to a slightly larger size than the fed controls. I would not hinge such a strong statement on this single observation.

*Reviewer #2 (Recommendations for the authors):*

The Dl staining in this article seems suboptimal, as the Dl signal should be clear membrane-associated rather than discrete puncta. Dl staining in the fly midgut can be tricky if fixed with 4% FAH while fixing with 100% methanol instead can solve this problem.

*Reviewer #3 (Recommendations for the authors):*

Some additional approaches could benefit the analysis.

(1) I think the model is that Vinc strengthens adhesion between ISC and EB to suppress EB differentiation. It would be interesting then to know whether EB and ISC physically separate more quickly in Vinc mutants, which could possibly be investigated through some live imaging.

(2) Would it be possible to do some epistasis tests to further examine the model that Vinc is required to maintain tension at AJ in the EB? eg What happens of the SqhDD is combined with Vinc mutants? Or activated Vinc with myosin loss-of-function?

(3) Can the authors provide evidence that tension is actually regulating Vinc localization at AJ in the EB. eg Is Vinc localization different in the experiments that increase or decrease myosin activity? What about with the a-cat mutants?

(4) Is Vinc required just for the strength of attachment to the ISC, or is it doing something else, eg could regulation of the Hippo pathway be relevant here?

[Editors’ note: further revisions were suggested prior to acceptance, as described below.]

Thank you for resubmitting your work entitled "Vinculin recruitment to α-catenin halts the differentiation and maturation of enterocyte progenitors to maintain homeostasis of the *Drosophila* intestine." for further consideration by *eLife*. Your revised article has been evaluated by Mone Zaidi (Deputy Editor) and a Reviewing Editor.

The manuscript has been improved but there are some remaining issues that need to be addressed, as outlined below:

There is a suggestion for a tweak to the discussion: Please can you try and address that, and we can then speedily accept?

*Reviewer #2 (Recommendations for the authors):*

I find no major concerns with the revised manuscript. However, one point would be helpful if added to the discussion section. The authors claimed that vinc non-autonomously regulates stem cell proliferation and suggest that EBs may suppress ISCs division through cadherin-mediated adhesion. Nevertheless, this model seems not quite consistent with the fact that MARCM clone of vinc mutant and RNAi shows no increase in clone size and reduces ISC proportion. The authors claim that some vinc clones lose ISCs (Dl staining), which means if one wait for a longer time the vinc LOF clones should be smaller than the control as the stem cells are lost. Therefore, I think the authors have not ruled out the possibility that vinc mutant EBs affect ISCs through certain long-range signals like upd or growth hormone, which may actually explain why vinc LOF MARCM clone does not grow bigger since long-range effects usually require global rather than local changes. Meanwhile, mechanical forces have also been found in the regulation of cell secretion in several cases.

*Reviewer #3 (Recommendations for the authors):*

The authors have satisfactorily addressed the issues raised in the initial review, and I have no further concerns with the manuscript.

---

## [Author Response]

Essential revisions:1) Explicitly examine whether vinculin in enteroblasts is regulated by changes in mechanical tension and/or cadherin-mediated adhesion.A pillar of the authors' model is that enteroblast vinculin exerts its effects through a mechanosensitive mechanism involving E-cad and a-cat. Work in other epithelial tissues has demonstrated that the E-cad-a-cat-vinc complex is mechanosensitive, but this mechanosensitivity has not been demonstrated in the fly gut. Since some canonical functions of E-cadherin are known to differ in the fly gut, it is important to establish that mechanosensitive regulation of vinc is conserved.The chief evidence the authors present for mechanosensitivity is that single genetic perturbations of vinc and myosin produce similar phenotypes. However, the phenotypic readouts that are shown (nuclear size, stem cell production, differentiation index, retention index) are indicators of stem cell renewal/differentiation. These types of effects could in principle arise through numerous mechanisms, some of which are not necessarily related to tension.Reviewer 3 suggests several approaches to more directly examine whether vinc in enteroblasts is responding to (or perhaps regulating) changes in mechanical tension. These include epistasis tests with vinc and loss- or gain-of function alleles of myosin, epistasis tests with vinc and E-cad (or perhaps E-cad mutants that lack the extracellular binding domain or the a-cat binding site), or examining the subcellular localization of vinc following loss- or gain-of-function myosin manipulations and/or in a-cat mutants. One of these suggestions, or an equivalent experiment, will be essential in a revised manuscript.

The reviewers question whether the E-cadherin/ a-catenin/ vinculin complex is mechanosensitive in the intestinal epithelium. To our knowledge, the cryptic vinculin binding site (VBS) on a-catenin only becomes accessible when a-catenin is stretched in a tension-dependent manner, as shown originally *in vitro* (Yonemura et al., 2010; Yao et al., 2014). The interaction between vinculin and a-catenin via the VBS was later confirmed in *Drosophila* (Sarpal et al., 2019; Alegot et al., 2019). Thus, our experiments in Figure 6E where the a-catenin RNAi phenotype is rescued by reintroduction of a full-length, but not a VBS-lacking a-catenin strongly suggest that vinculin is recruited to adherens junctions by this well-defined mechanism of binding to force-stretched a-catenin, which in turn impacts tissue turnover.

Nonetheless, we sought to provide additional evidence to support the mechanosensitive regulation of tissue turnover by vinculin, as requested. We tested the epistatic relationship between myosin II and vinculin, by expressing a constitutively active form of myosin II regulatory light chain (Sqh^DD^) in all progenitor cells in a *vinculin* mutant background. We observed rates of differentiation similar to those observed in *vinc^102.1^* guts (new Figure 7A-B), suggesting that vinculin acts downstream of myosin II induced tension. Furthermore, we obtained data showing that vinculin recruitment to adherens junctions increased with tension in EBs (new Figure 7C-D). Thus, these experiments demonstrate that vinculin is acting in the intestine as a mechanoeffector, in a similar manner to that characterized in other epithelia (described pages 10-11 in section: ‘Vinculin links cellular tension to EB differentiation’).

2) Provide textual explanations and/or experimental evidence to clarify certain cell-specific phenotypes.All reviewers felt that interpretations of some cell type-specific genetic manipulations were either oversimplified or, potentially, contradictory:a. There was concern that some experimental outcomes (MARCM expts, knockdown experiments using ISC+EB vs EB drivers ) do not support the authors' conclusion that vinc acts in enteroblasts and not in stem cells. Can the authors clarify these apparent discrepancies through either textual discussion or new data?

We apologise that we were unclear in the text. With hindsight, the confusion may have been caused by our describing the phenotype of MARCM clones before reporting the accumulation of EBs in the *vinc^102.1^* guts. Therefore, we swapped these two sections and improved the description of these experiments in the manuscript (see section: “The pool of enterocyte progenitors expands upon vinculin depletion” pages 6-8).

In brief, we do not think that our results are at odds with the phenotype of vinculin knockdown using the ISC and ISC/EB drivers – we realise the text was misleading and hope to have clarified our observations in the revised manuscript (pages 7 and 8). From cell conditional RNAi experiments, like the reviewer, we would predict that vinculin knockdown or loss of function in mitotic clones (MARCM experiments, Figure 4E-G) will induce accelerated differentiation of vinculin deficient enteroblasts, which in turn will increase proliferation. We observed that *vinc^102.1^* or *vinc RNAi* mitotic clones contained similar number of cells compared to control clones, but reduced proportion of stem cells (Figure 4G). We interpret this as indicating that to maintain an equivalent clone size, stem cells must have divided more frequently, with some divisions producing two differentiated daughter cells. This type of symmetric division would increase the EB pool (as seen in Figure 4—figure supplement 2B), at the expense of the ISC population, in turn decreasing long term clonal growth potential. Altogether, the results obtained with MARCM clones highlight changes in tissue dynamics compatible with those observed with cell-specific vinculin knockdowns.

b. The conclusion that progenitor cell vinc does not act through integrins, which is presented as a key finding, is based on the result that mys or talin RNAi does not phenocopy vinc RNAi. However, these data do not exclude a scenario in which vinc acts through both cadherin and integrins, with the cadherin pathway dominating. Either additional data should be provided that directly test the involvement of integrins, or the text should be modified to discuss the limitations of the current approach.

The reviewer raised an important point. To test this we had to overcome the ISC defect of *mys* or *talin* RNAi, and specifically tested their function in enteroblasts using the KluGal4 driver. This revealed a similar phenotype of accelerated differentiation, assayed with the ReDDM system (see new Figure 6 —figure supplement 4). Thus, as the reviewer suggested both integrins and cadherins function in this process, we have amended the text to indicate this (see page 10, and sentence in the discussion page 12). It appears however that, unlike vinculin, they also have a key role in ISCs.

c. Why does vinc::RFP rescue the vinc null while vincCO has little effect?

As shown in Figure 7, figure supplement 2, the accelerated turnover seen in *vinc^102.1^* guts is attenuated by expression of Vinc^CO^. However, Vinc^CO^ expression does not affect the little turnover in the guts of *yw* males. Previous work has shown that Vinc^CO^ has numerous effects, none of which appear to be a simple gain of normal activity (Maartens et al., 2016) which we highlighted in the text (see page 11).

3) As shown, the starvation/refeeding and survival assays are difficult to interpret because their use of whole-animal vinc nulls means that the outcomes cannot necessarily be ascribed to enteroblast vinculin. These experiments should either be repeated in a manner that reveals the contribution of enteroblast vinculin, or else removed from the study (as suggested by Reviewer 1).

We have now directly tested if depletion of vinculin from intestinal progenitor cells affects organismal survival in starvation/refeeding experiments by expressing vincRNAi under the control of esg-Gal4 in adults. Similar to our observations with the *vinc^102.1^* flies (which have now been moved to Figure 8—figure supplement 1), we concluded that *vinculin* knockdown in progenitor cells favours fly survival after a period of starvation (see Figure 8 A-B).

Whilst we cannot exclude that vinculin may play important roles in other tissues to regulate organismal physiology throughout development and adulthood (as we now note in the text, page 12), our result shows an important contribution of vinculin in esgGal4+ cells to adult fly survival. We have rephrased our conclusions accordingly (page 12).

4) Dl immunostaining – Can the authors comment on how quantitation of Dl+ cells was performed with the punctate immunostaining pattern evident in, for instance, Figure 4a-b? In online discussion among the reviewers, there were differing opinions about Dl staining patterns but agreement that ascribing Dl puncta to individual cells may be unreliable without a membrane marker.

Dl+ cells are quantified manually, on z-stack reconstructions spanning half the depth of the intestine from samples processed side by side with control guts. We routinely identify DAPI+Dl+ cells as small cells, located near the basal side of the epithelium. In Figure 4A-B, Dl+ staining is mutually exclusive with the b-Galactosidase staining marking Notch activating enteroblasts, confirming the effective use of this marker. We have now included this description in the method section ‘quantifications and statistics’ (page 19).

As a side note, in our experience, with our paraformaldehyde fixation method (described in material and methods), the Dl staining is mostly in puncta, but can, in some experimental conditions, appear strongly enriched at the membrane. Thus, while we appreciate the concern of the reviewer that the punctate pattern makes it harder to assign to a particular cell, in practise it is not difficult, and is standard in the field (e.g. https://doi.org/10.1016/j.celrep.2019.08.014 (Jun-Kit Hu and Jasper, Cell reports, 2019) Willms et al., Cell Reports, 2020 https://doi.org/10.1016/j.celrep.2020.108400 Loza-Coll et al., EMBO J., 2014 https://doi.org/10.15252/embj.201489050 good example of membrane and dotty within a same panel).

Reviewer #1 (Recommendations for the authors):I am mostly repeating myself here but the experiments I would suggest are:– Repeating some key experiments while recording additional, more standardised metrics, in particular evaluating PH3 and rates of differentiation with ISC-Gal4, esg-Gal4, klu-Gal4 and MARCM clones with UAS-vincRNAi.

We have improved the consistency of the measurements by adding measures of cell density throughout, as a proxy for cell proliferation and differentiation, and in cases where we could see a phenotype, we have added the mitotic index and rate of differentiation.

– Establishing the autonomy of the phenotype in vinc[102.1] in one of the ways suggested above.

We have repeated the survival experiment, this time knocking down vinculin in esg+ cells in the adult (Figure 8B and text page 12).

– Establishing the autonomy of the α-catenin phenotype in ISCs and EBs, and whether the full deletion of the M1 region fails to rescue the phenotype.

As discussed, we do not expect the autonomy of a-Catenin to be the same as vinculin. For technical reasons detailed above, we did not repeat these experiments with an alternative construct with the full deletion of the M1 region.

– Establish whether expression of VincCO (or VincFL) can change the behaviour of the system when forced to regenerate.

We feel this is outside the scope of this manuscript.

Some technical fastidious stuff, sorry:– Lifespan experiment – individuals dissected for gut staining cannot be assumed to experience the same risk as the rest – the correct approach is to remove them from the cohort or censor them, otherwise the amount of animals at risk after dissection is not the right one.

We have removed from the cohort the vials containing flies that were due to be dissected during the course of the experiment. The new corresponding graph is included in Figure 8—figure supplement 1E and the data included in the associated data source. Note that we stopped the census at day 18 (as tubes subsequently were used for dissections). The method section (page 19) has also been amended.

– Some language mishaps:There is a couple of instances where esgTS-FO patches are referred to as "clones", it should be corrected.The authors refer to 29ºC in experiments using tub-Gal80ts as the 'permissive' temperature, which is not the usual way to refer to this in the literature, as this temperature is when Gal80ts stops functioning – I understand that for the activity of Gal4 this is 'permissive' but the primary molecular event is the destabilisation of Gal80ts and that is how the community is used to think of this – if the authors wish to maintain this nomenclature, perhaps some more explicit mention of what they mean would be useful in the main text.

This has been corrected in figure legends (Figure 2; Figure 2—figure supplement 2, and figure 7—figure supplement 2). An additional explanation was added in the methods (page 16), and the text was modified in the figure legends changing ‘14 days after clone induction (ACI)’ to ‘14 days after induction of Gal4 expression’.

The use of the term "compensatory proliferation" is a bit out of context here, as it has been employed traditionally to refer to proliferation to compensate apoptosis; however, in this paper it is argued that the proliferation arises from loss of contact inhibition due to differentiation, so the phenomenology does not parallel the traditional use of the term in *Drosophila*.

"compensatory proliferation" has been changed to ‘increased ISC proliferation’ (see page 11)

The authors correctly conclude that flies are more resistant to starvation and they word it very similarly in the main text, but in the abstract they phrase this observation as "Improved recovery after starvation", which is misleading as the phenotype is circumscribed to the starvation period: the survival curve is essentially the same AFTER the re-feeding, and the only difference in the re-feeding period is this single time point when the guts from vinc[102.1] guts recovered to a slightly larger size than the fed controls. I would not hinge such a strong statement on this single observation.

The text was changed in the abstract from “Removing vinculin increases enteroblast differentiation and numbers, resulting in an enlarged gut with improved ability to recover after starvation” to “Removing vinculin increases enteroblast differentiation and numbers, resulting in an enlarged gut with improved resistance to starvation”.

In the result section (page 11), we removed the sentence describing the larger gut size and focused on survival, but kept a sentence in the discussion (page 13).

Reviewer #2 (Recommendations for the authors):The Delta (Dl) staining in this article seems suboptimal, as the Dl signal should be clear membrane-associated rather than discrete puncta. Dl staining in the fly midgut can be tricky if fixed with 4% FAH while fixing with 100% methanol instead can solve this problem.

As mentioned above (point 4, essential revisions) we find Dl staining variable, and despite the punctate appearance we are confident of our our quantification, as verified by the mutually exclusive staining of Dl+ in ISCs and LacZ+ in EBs, as shown in Figure 4A-B. We will try methanol fixation in future experiments.

Reviewer #3 (Recommendations for the authors):Some additional approaches could benefit the analysis.1) I think the model is that Vinc strengthens adhesion between ISC and EB to suppress EB differentiation. It would be interesting then to know whether EB and ISC physically separate more quickly in Vinc mutants, which could possibly be investigated through some live imaging.

Unfortunately, a live imaging approach like this one is not easily done with our current tools and imaging set-up. We have attempted to measure the distance between Dl+ ISC and Su(H)lacZ+ EBs on fixed samples but found large variations both in control and *vinc^102.1^* guts, probably representative of cells at different stages of activation. In any case, absence of changes in total a-Catenin staining in *vinc RNAi* clones (Figure 6—figure supplement 1) suggests that the integrity of adherens junctions is maintained between cells, even in the absence of vinculin. It is therefore possible that the changes occurring between ISCs and EBs in *vinc^102.1^* mutants reflect changes in cellular tension, and changes in the engagement of adherens junction proteins with the actin cytoskeleton rather than complete separation. Thus new tools will be required to elucidate the role of vinculin in this process.

2) Would it be possible to do some epistasis tests to further examine the model that Vinc is required to maintain tension at AJ in the EB? eg What happens of the SqhDD is combined with Vinc mutants? Or activated Vinc with myosin loss-of-function?

Please refer to point 1, essential revisions above – our new data suggest that vinculin acts a mechanoeffector downstream of myosin II induced tension (new Figure 7A-B).

3) Can the authors provide evidence that tension is actually regulating Vinc localization at AJ in the EB. eg Is Vinc localization different in the experiments that increase or decrease myosin activity? What about with the a-cat mutants?

Please refer to point 1, essential revisions above – we observed increased vinculin recruitment to adherens junctions when a constitutively active form of myosin II was expressed in EBs (new Figure 7C-D). Conversely, vinculin recruitment to EB adherens junctions is reduced upon a-catenin RNAi expression (new Figure 6—figure supplement 1B,C).

4) Is Vinc required just for the strength of attachment to the ISC, or is it doing something else, eg could regulation of the Hippo pathway be relevant here?

The experimental evidence points to a role of vinculin at cell junctions, but we cannot exclude other roles. Our efforts with Hippo pathway reporters have so far been inconclusive, and so this question requires further work, which is beyond the scope of this manuscript.

We are excited that our study has identified a mechanosensing role of vinculin at cell junctions. This is a starting point for further evaluation of conserved roles of vinculin dependent mechanotransduction in intestinal homeostasis and regeneration.

[Editors’ note: further revisions were suggested prior to acceptance, as described below.]

Reviewer #2 (Recommendations for the authors):I find no major concerns with the revised manuscript. However, one point would be helpful if added to the discussion section. The authors claimed that vinc non-autonomously regulates stem cell proliferation and suggest that EBs may suppress ISCs division through cadherin-mediated adhesion. Nevertheless, this model seems not quite consistent with the fact that MARCM clone of vinc mutant and RNAi shows no increase in clone size and reduces ISC proportion. The authors claim that some vinc clones lose ISCs (Dl staining), which means if one wait for a longer time the vinc LOF clones should be smaller than the control as the stem cells are lost. Therefore, I think the authors have not ruled out the possibility that vinc mutant EBs affect ISCs through certain long-range signals like upd or growth hormone, which may actually explain why vinc LOF MARCM clone does not grow bigger since long-range effects usually require global rather than local changes. Meanwhile, mechanical forces have also been found in the regulation of cell secretion in several cases.

We thank the editors and reviewers for taking the time to assess our revised manuscript. We have now added a sentence in the discussion to acknowledge that we cannot exclude a contribution of vinculin activity to long range signalling.

Page 13: sentence added: “Whilst our data mostly support a direct role of vinculin at cell junctions in EBs, we cannot exclude that vinculin activity might also impact long range signals regulating stem cell proliferation, especially in light of the mild effects observed when vinculin is depleted in small clones of cells rather than globally (Figure 4E-G).”

The revised version (containing tracked changes) has been uploaded in place of the current version. We also noted that 3 figures (Fig6, Figure 6S1 and Figure 8S1) were low resolution and appeared pixelated, we have thus replaced the jpgs with better quality ones.